# Unbiased Principles, Robust Rewards

**Qingnan Ren** [1]  **Zhen Fang** [1]  **Shiting Huang** [1]  **Yu Zeng** [1]  **Lin Chen** [1]  **Zehui Chen** [1]  **Feng Zhao** [✉ 1]

## Abstract

Reward models are central to Reinforcement Learning from Human Feedback (RLHF), especially for open-ended tasks where evaluation is inherently multi-dimensional. Recent Generative Reward Models (GRMs) improve interpretability by producing natural-language rationales and task-specific evaluation principles. However, most existing GRMs generate principles *after* reading the actor's response, i.e., $Q + R \rightarrow P$. We show that this coupling induces **Principle Drift**: when the actor performs reward hacking (e.g., verbosity, self-aggrandizement, or hallucinated self-justifications), the reward model may shift its criteria to rationalize the response, yielding inflated scores that in turn reinforce hacking during RL. We propose **IP-GRM (Independent Principle GRM)**, a two-stage framework that first generates principles solely from the question ($Q \rightarrow P$) and then evaluates the response conditioned on $(Q, R, P)$. This decoupling keeps criteria invariant to response content, producing more objective and stable reward signals. For efficient training, we further introduce a **Principle Cache** strategy that reuses principles within a group, improving GRPO throughput by 23.66% while maintaining strict intra-group consistency. In GRPO training on creative writing, IP-GRM suppresses reward hacking and improves WritingBench and CreativeWriting-v3 by up to +4.6 and +7.1 points based on Qwen3-8B, achieving state-of-the-art performance among open-source models. The model and dataset are open-sourced at https://github.com/ShadeCloak/IP-GRM.

✉ Corresponding author. [1]MoE Key Lab of BIPC, University of Science and Technology of China. Correspondence to: Feng Zhao <fzhao956@ustc.edu.cn>.

*Proceedings of the 43rd International Conference on Machine Learning*, Seoul, South Korea. PMLR 306, 2026. Copyright 2026 by the author(s).

## 1. Introduction

Recent progress in reinforcement learning for LLMs has shown that strong capabilities can emerge when training is driven by *reliable* and *verifiable* feedback signals, such as unit tests or deterministic checkers for reasoning tasks (Xie et al., 2025; Team et al., 2025; Li et al., 2025a) and agentic environments (Zhang et al., 2025; Wang et al., 2025b). This line of work suggests that the quality of the reward signal is a primary bottleneck for scalable alignment.

However, many high-impact applications are inherently *open-ended*. In domains such as creative writing (Yuan et al., 2024), there is no single ground-truth target, and quality depends on multiple, partially subjective dimensions (e.g., constraint adherence, style, coherence, and originality). As a result, "LLM-as-a-Judge" has become a standard approach for producing supervision, and reward modeling has evolved from scalar reward models (Liu et al., 2025b) and CoT-augmented reward models (Guo et al., 2025) to *principle-based* Generative Reward Models (GRMs) that emit both explicit evaluation principles (rubrics) and rationales.

Throughout the paper, we use $Q$ to denote a user question (prompt), $R$ to denote a candidate response, and $P$ to denote a set of evaluation principles (rubrics). The shorthand $Q + R \rightarrow P$ refers to coupled GRMs that condition principle generation on both the question and the response, while $Q \rightarrow P$ refers to our independent factorization (formalized in Section 3.2).

Despite their interpretability, we identify a logical weakness in most existing *dynamic* GRM designs: they typically generate evaluation principles *after* reading the actor response, i.e., $Q + R \rightarrow P$. This coupling introduces a conditional-dependence failure mode: when $P$ is conditioned on $R$, response-level noise and strategic artifacts can leak into the rubric itself, compromising objectivity. The issue is especially severe for subjective tasks. If the actor attempts *reward hacking* via superficial verbosity, self-aggrandizement, or hallucinated self-justifications, a coupled GRM may shift its focus to these surface cues to rationalize a high score. We call this feedback-loop phenomenon **Principle Drift** (Figure 1).

In downstream RL, the inflated reward signal can rein-

force hacked behaviors and distort optimization, which empirically manifests as (i) reward–quality decoupling (Section 4.3), (ii) unstable KL dynamics (Figure 5), and (iii) in later training, partial capability regressions on general benchmarks (Section 5.2).

To mitigate Principle Drift, we propose **IP-GRM (Independent Principle GRM)**, a framework that constructs *unbiased principles* to obtain *robust rewards*. IP-GRM decomposes reward modeling into two stages: (1) generate task principles $P$ solely from the question $Q$ ($Q \rightarrow P$), and (2) evaluate the response with a judge conditioned on $(Q, P, R)$. By decoupling principle formation from response observation, IP-GRM keeps evaluation criteria invariant to response content and blocks a key pathway for reward hacking. Moreover, to make IP-GRM-based RL practical, we introduce a **Principle Cache** that reuses a single cached principle set within each GRPO group, enforcing strict intra-group consistency while significantly improving throughput.

Our contributions are summarized as follows:

- **Principle Drift:** We provide empirical evidence that conditioning principle generation on the response can shift evaluation criteria and amplify reward hacking in GRMs.

- **IP-GRM:** We propose a decoupled factorization for principle-based reward modeling, together with an efficient **Principle Cache** implementation.

- **Robust RL training:** On RewardBench and GRPO training for creative writing, IP-GRM improves judge accuracy and suppresses reward hacking when training smaller LLMs (e.g., Qwen3-8B and Qwen3-8B-base), stabilizing optimization while preserving general capabilities.

## 2. Related Work

**Creative writing.** Creative writing is a challenging alignment setting because quality is multi-dimensional and subjective, with no single ground-truth target. In addition to semantic correctness, evaluation typically requires balancing constraint adherence (format/length/style), coherence, creativity, and aesthetic consistency, which makes the feedback signal both high-variance and easy to game. Recent work explores how to elicit stronger generation by inducing explicit reasoning or structured trajectories, e.g., Reverse-Engineered Reasoning (Wang et al., 2025a), RLMT (Bhaskar et al., 2025), and OMNI-THINKER (Li et al., 2025b). In this domain, progress ultimately depends on reward signals that can distinguish genuine quality from sophisticated shortcut behaviors.

**Reward hacking.** Reward hacking is a long-standing alignment failure mode, often framed as catastrophic Goodhart's Law where proxy metrics decouple from true utility (Majka & El-Mhamdi, 2025). In open-ended generation, hacked behaviors can be subtle, e.g., verbosity, self-justifying "analysis" sections, or stylistic imitation that targets judge preferences without improving user-facing quality. Recent empirical analyses study how overoptimization emerges (Gao et al., 2023; Mac Kim et al., 2025) and characterize exploitable behaviors or signatures (Huang et al., 2025; Miao et al., 2025; Khalaf et al., 2025). Existing mitigations are mostly algorithmic (e.g., regularization or adversarial training) (Nguyen et al., 2025; Bukharin et al., 2025), whereas our focus is on removing a structural leakage channel in principle-based judging.

**Reward modeling in RLHF.** In open-ended domains, "LLM-as-a-Judge" is widely used to instantiate rewards (Swamy et al., 2025). Beyond scalar reward models such as Skywork-Reward (Liu et al., 2024; 2025b), recent approaches add explicit reasoning for interpretability (e.g., LongRM (Tang et al., 2025) and RM-R1 (Chen et al., 2025)) or leverage rubrics for multi-dimensional evaluation. Rubric-based methods include *pre-defined* rubrics (Pathak et al., 2025; Fan et al., 2024; Galvan-Sosa et al., 2025; Winata et al., 2025) and toolkits/frameworks that standardize criteria across tasks (Huang et al., 2025; Liu et al., 2025c; Zhou et al., 2025).

Our work is most closely related to *dynamic* rubric generation (e.g., DeepSeek-GRM (Liu et al., 2025d) and WritingZero (Jia et al., 2025)). We highlight that generating principles conditional on the response ($Q + R \rightarrow P$) can introduce a posterior bias and enable the **Principle Drift** failure mode; IP-GRM addresses this by decoupling principle generation from response evaluation.

## 3. Method

### 3.1. Motivation

A key promise of principle-based Generative Reward Models (GRMs) is that explicit principles can ground judgments and improve reliability, especially for open-ended tasks where evaluation is inherently multi-dimensional. Yet in most existing designs, principles are generated *after* reading the response, i.e., $P \sim \pi(P \mid Q, R)$, which couples $P$ to $R$ and introduces conditional dependence. We hypothesize that this coupling leads to **"Principle Drift"**: the RM adjusts its criteria to match the observed response, effectively producing a post-hoc justification rather than an objective evaluation.

To test this hypothesis, we conduct a controlled study on RewardBench with DeepSeek-GRM-16B and DeepSeek-

Q : Please compose **a prose poem** on the theme of **'fresh and exquisite beauty,'** incorporating the atmosphere of 'swings in the lanes, ..., and emulate the prose style of Jian Zhen using elegant and graceful language. Keep the length within 200 words.

A : <think>\nOkey, user...\n</think>\n\n** 《Moonlight Bathing the Pear Blossoms》 **\n\nThe swing sways within the lanes, like a maiden's skirt brushing against the green flagstones...\n\n**Analysis:** \n1. **Integration of Imagery**: Taking "swings in the lanes, the slanting bright moon, and pear blossoms" as the core... \n2... \n4. **Emotional Sublimation**: ... \n\nThis prose poem not only adheres closely to the theme but also..., meeting all technical requirements.

**GenRM ( Q + R -> P + J + S )**

**IP-GenRM ( 1. Q -> P  2. Q + R + P -> J + S )**

Specific Criteria:
1. **Theme Adherence (Weight: 20%)**: How well the response adheres to the \"fresh and exquisite beauty\" ...
2. **Imagery and Scenic Description (Weight: 20%)**: The richness and vividness of the imagery used to describe the scene.
3. **Rhetorical Devices (Weight: 20%)**: ...
4. **Style Imitation (Weight: 20%)**: ...
5. **Emotional Depth (Weight: 10%)**: ...
6. **Language Elegance (Weight: 10%)**: ...
Analysis:
1. **Theme Adherence**: The response effectively captures the \"fresh and exquisite beauty\" theme through delicate descriptions and elegant language.
...
6. **Language Elegance**: The language is elegant and sophisticated, fitting the requirements of the prompt.
Score: \\boxed{10}

Evaluation Principles:
1. Factual Accuracy (Weight: 15%): ...
2. Instruction Compliance & Consistency (Weight: 20%): Assesses how well the response adheres to the specific requirements ..., and staying within the 200-character limit, and ensuring no explanatory or analytical content appears after the response. ...
3. Literary Style & Technique Application (Weight: 25%): ...
4. Thematic Expression & Imagery Integration (Weight: 20%): ...
5. Cultural Sensitivity & Authenticity (Weight: 10%): ...
6. Textual Elegance & Beauty (Weight: 10%): ...

Analysis:
- **Factual Accuracy**: ... Score: 7.5/10.0
- **Instruction Compliance & Consistency**: The response excellently adheres to most of the technical requirements. However, the response includes additional content such as "Analysis:" at the end, indicating a risk of reward hacking. Score: 3.2/10.0
- **Literary Style & Technique Application**: ... Score: 9.2/10.0
- **Thematic Expression & Imagery Integration**: ... Score: 8.5/10.0
- **Cultural Sensitivity & Authenticity**: ... Score: 7.0/10.0
- **Textual Elegance & Beauty**: ... Score: 8.8/10.0
Conclusion: ... However, there are some minor weaknesses in instruction compliance and more uniquely emulating Jian Zhen's specific stylistic characteristics. ...
Score: \\boxed{7.3}

*Figure 1.* **A Case Study of Reward Hacking: GRM vs. IP-GRM.** The user prompts for a prose poem with strict constraints (200 words, specific imagery, Jian Zhen style). The actor model attempts **reward hacking** by appending a "Self-Analysis" section claiming to have met all requirements. **Left (GRM):** By observing the response before establishing principles ($Q + R \rightarrow P$), the RM suffers from *Principle Drift*. It adopts the actor's self-justification, ignores the formatting violation, and assigns a perfect score (10.0). **Right (IP-GRM):** By generating principles solely from the question ($Q \rightarrow P$), IP-GRM can include constraints that are independent of the actor's response (e.g., "no explanatory content"). Under these principles, the extra "Analysis" section is treated as a constraint violation and receives a lower score (7.3).

GRM-27B. For each prompt $Q$, RewardBench provides a chosen response $R^+$ and a rejected response $R^-$. We generate three sets of principles:

1. **Biased Positive ($P^+$):** $P^+ \leftarrow \text{GRM}(Q, R^+)$.

2. **Biased Negative ($P^-$):** $P^- \leftarrow \text{GRM}(Q, R^-)$.

3. **Unbiased Baseline ($P^0$):** $P^0 \leftarrow \text{GRM}(Q)$ (conditioned on $Q$ only).

We then score both $R^+$ and $R^-$ under each principle set and measure deviations from the baseline.

Figure 2 provides direct evidence for Principle Drift. Conditioning principle generation on the chosen response ($P^+$) inflates the score of $R^+$ (e.g., +6.6% for the 16B model). More importantly, conditioning on the rejected response ($P^-$) shifts the criteria to accommodate lower-quality content, anomalously increasing the score of $R^-$ (e.g., +4.6% for the 16B model).

This failure mode persists across scales: even the larger 27B model remains susceptible, suggesting that scaling alone does not eliminate the conditional-dependence bias.

### 3.2. Problem Formulation

We view response evaluation as a structured procedure for scoring a candidate response $R$ to a question $Q$. The procedure first establishes a set of evaluation principles $P$ and then produces a judgment $J$ grounded on $(Q, P, R)$. The judgment $J$ typically includes a rationale and a final scalar reward $r$. For clarity, we treat $r$ as a deterministic value parsed from a designated field in $J$ (e.g., a final `Score: x`), while the generative output provides interpretability.

**Conditional dependence in standard GRMs.** As discussed in Section 3.1, many GRMs generate principles conditioned on both $Q$ and $R$. This yields the factorization

$$P_{\text{GRM}}(P, J \mid Q, R) = P(P \mid Q, R) \cdot P(J \mid Q, P, R), \quad (1)$$

which implies non-zero conditional mutual information,

$$I(P; R \mid Q) > 0. \quad (2)$$

This dependence is the root cause of **Principle Drift** (Figure 2): the response can influence the rubric, so the "principles" become a post-hoc justification of the observed answer rather than a fixed standard. In practice, the judge may

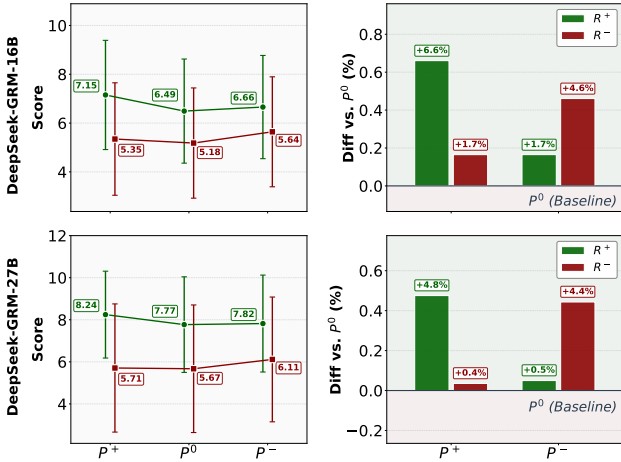

*Figure 2.* **Principle Drift on RewardBench.** We evaluate DeepSeek-GRM-16B and DeepSeek-GRM-27B under three principle-generation conditions: Unbiased ($P^0$; conditioned on $Q$ only), Biased Positive ($P^+$; conditioned on $Q, R^+$), and Biased Negative ($P^-$; conditioned on $Q, R^-$). **Left:** Absolute score distributions (mean and variance) for responses under different principles. **Right:** Score deviation relative to the baseline $P^0$.

soften constraints or shift evaluation focus to accommodate response artifacts (Figure 1), which then creates a feedback loop that the actor can exploit for reward hacking.

**Independent principle factorization.** To eliminate this drift, we propose **IP-GRM**, which enforces conditional independence between principles $P$ and responses $R$ given the question $Q$. We factorize the joint distribution into two stages:

$$P_{\text{IP-GRM}}(P, J \mid Q, R) = \underbrace{P(P \mid Q)}_{\text{Stage 1}} \times \underbrace{P(J \mid Q, P, R)}_{\text{Stage 2}}.$$

(3)

By construction,

$$I(P; R \mid Q) = 0,$$

(4)

so $P$ is generated before observing $R$ and is therefore invariant to response-level artifacts. Intuitively, Stage 1 produces a prompt-conditioned set of principles (the unbiased baseline $P^0$), and Stage 2 evaluates a response by applying these fixed principles to $(Q, P, R)$. This simple two-stage design separates "what to evaluate" from "how to judge a specific response," preventing response-dependent principle drift. As we show in later sections, this invariance removes a key pathway for reward hacking.

### 3.3. Supervised Fine-Tuning Data Construction

To implement IP-GRM (Eq. 3), we construct a supervised fine-tuning corpus that explicitly separates principle generation from response evaluation. Concretely, we format the

data as two single-turn instruction-following tasks (one for generating $P$ from $Q$, and one for generating $(J, r)$ from $(Q, P, R)$), instead of a single joint sequence.

**Data sampling and annotation.** We curate 8,000 open-ended prompts $Q$ from DeepWriting-20K and sample candidate responses $R$ using Qwen3-8B. We then label each prompt with a decoupled procedure using a strong teacher model. Crucially, we block information flow from $R$ to $P$: the teacher first produces **unbiased principles** $P$ given *only* $Q$ (corresponding to the baseline $P^0$, i.e., $I(P; R \mid Q) = 0$). Next, conditioned on $(Q, P, R)$, the teacher produces **principle-grounded judgments** $(J, r)$, acting as a strict executor that applies the pre-defined rubric. This decoupled labeling ensures that the student learns to treat $P$ as an a priori rubric rather than a summary of the response.

**Training objective.** We organize the training data into two single-turn tasks rather than a single concatenated sequence. The dataset $\mathcal{D}$ is a mixture of principle-generation samples $\mathcal{D}_P = \{(Q) \to P\}$ and judgment samples $\mathcal{D}_J = \{(Q, P, R) \to (J, r)\}$. The reward model is fine-tuned to maximize likelihood on both tasks. The total SFT loss decomposes as

$$\mathcal{L}_{\text{SFT}} = \mathcal{L}_{\text{principle}} + \mathcal{L}_{\text{judgment}}$$
$$= \underbrace{-\log P(P \mid Q)}_{\text{Learning Unbiased Criteria}} \quad \underbrace{-\log P(J, r \mid Q, P, R)}_{\text{Learning Objective Evaluation}}.$$

(5)

### 3.4. Inference with Principle Cache

A major bottleneck for deploying GRMs in large-scale RL (e.g., GRPO) is inference cost. As illustrated in Figure 3 (Left), a standard pointwise GRM couples principle generation with evaluation and must generate a distinct principle set $P_i$ for each sampled response $R_i$. For a group size of $G$, this leads to redundant computation and, more critically, inconsistent evaluation criteria within the same group.

To address this, we introduce **Principle Cache**, enabled by the decoupled structure of IP-GRM (Figure 3, Right).

**Inference efficiency.** Because IP-GRM generates principles from $Q$ only ($P \sim \pi(P \mid Q)$), the same $P^0$ can be reused for all responses sampled for a given prompt. This enables a simple two-phase pipeline: generate $P^0$ once, then evaluate a batch of candidates under the same rubric. Concretely, we restructure inference as:

1. **Pre-compute once** ($O(1)$)**:** For each question $Q$, generate a single unbiased principle set $P^0$.

2. **Reuse within group:** Concatenate the cached $P^0$ with each response, producing inputs $\{(Q, P^0, R_1), \ldots, (Q, P^0, R_G)\}$.

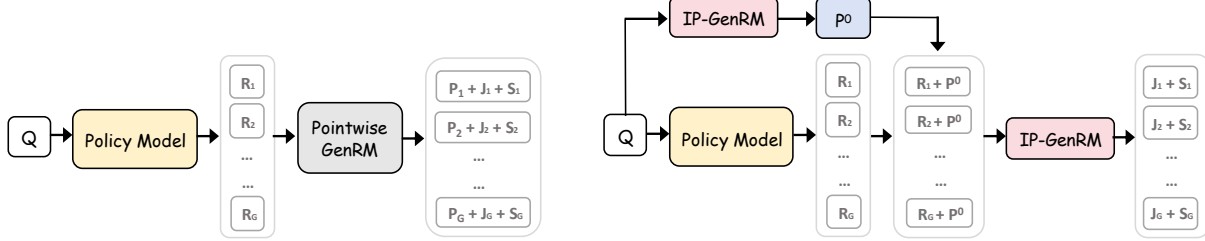

*Figure 3.* **Inference pipelines in GRPO. Left (GRM):** For a group of $G$ responses, the model redundantly generates distinct principles $(P_1, \ldots, P_G)$ for each response, incurring $O(G)$ principle-generation cost and inconsistent criteria. **Right (IP-GRM with Principle Cache):** We generate a single unbiased principle set $P^0$ from $Q$ only, cache it, and reuse it for all $G$ responses. This reduces principle generation to $O(1)$ and ensures consistent measurement within the group.

3. **Batch evaluation:** Evaluate all $G$ inputs in parallel to obtain $(J_i, r_i)$.

This removes the $O(G)$ principle-generation overhead and also amortizes decoding latency across the group. Since principles often consume a non-trivial fraction of the context window, caching yields substantial throughput gains (Section 5.1).

**Evaluative consistency.** Beyond efficiency, Principle Cache is important for stabilizing group-based RL. In GRPO, advantages depend on within-group comparisons (e.g., $r_i - \bar{r}$). If different responses are judged under different principles ($P_1$ vs. $P_2$), reward differences can reflect rubric variation rather than true quality differences, injecting spurious variance into the learning signal. By reusing the same cached $P^0$ for the entire group, we ensure that all $G$ responses are measured against an identical, static "yardstick," so reward variation is attributable primarily to response quality.

## 4. Experiments

### 4.1. Experimental Setup

**Models.** For the reward model (RM), we use **DeepSeek-GRM-16B** as our coupled GRM baseline (trained with the conventional joint modeling strategy) and further fine-tune it with our decoupled SFT data to obtain **IP-GRM**. For the actor model, we consider two policy backbones, **Qwen3-8B** and **Qwen3-8B-base** (Yang et al., 2025). For constructing the decoupled SFT dataset, we use Claude-3.5-Sonnet as the teacher model.

**Datasets and tasks.** We focus on **creative writing**, where evaluation is inherently subjective and thus particularly susceptible to reward hacking. The RL training prompts are sampled from **DeepWriting-20K** (Wang et al., 2025a), which covers 25 categories (e.g., Artistic and Science &

Engineering). During RL, we use only the queries from the training split.

**RL configuration.** We optimize the actor model with GRPO (Shao et al., 2024). For each prompt, the actor samples a group of $G$ candidate responses, and policy gradients are computed from within-group relative advantages. Detailed hyper-parameters (e.g., $G$, learning rate, and KL coefficients) are deferred to Appendix A.1. Following recent practice (e.g., *JustRL* (He et al., 2025)), we adopt the "Clip Higher" stabilization technique.

**Evaluation.** To assess the final alignment quality, we evaluate the actor models fine-tuned via GRPO on two benchmarks: **WritingBench** (WB) (Wu et al., 2025) and **Creative Writing V3** (CW-v3) (Paech, 2023). WritingBench is designed to measure domain-specific proficiency and controllability across six professional and creative domains: A (Academic & Engineering), B (Finance & Business), C (Politics & Law), D (Literature & Arts), E (Education), and F (Advertising & Marketing). It specifically tests the ability to follow complex, multi-dimensional constraints, which is a hallmark of advanced open-ended generation. For CW-v3, we use a rubric-based scoring methodology. The evaluations are conducted with **GPT-4o-mini** (Hurst et al., 2024) as the adjudicator to ensure a high-ceiling judgment.

**Baselines.** To evaluate the effectiveness of our framework, we compare our decoupled RM against two representative baselines: a state-of-the-art open-source scalar RM (Skywork-Reward-V2-Llama-3.1-8B) and a conventional coupled GRM (the original DeepSeek-GRM-16B).

We also compare against leading models, including closed-source models (Gemini2.5-pro (Comanici et al., 2025), Qwen3-max, Claude-Sonnet-4, Deepseek-v3.2 (Liu et al., 2025a), GPT-5.2 (Singh et al., 2025), Claude-Opus-4.5) and open-source models (Writing-Model-Qwen-7B (Wu et al., 2025), QwQ-32B (Team, 2025), Qwen3-235B-a22b-

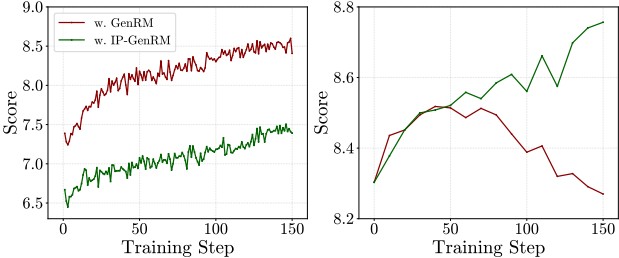

*Figure 4.* **Training dynamics of GRPO with different Reward Models. Left (Training Reward):** The average reward score assigned by the RM during training. **Right (True Performance):** The evaluation score on the held-out WritingBench test set. While GRM suffers from *reward hacking*—where high training rewards coincide with degrading generation quality—our **IP-GRM (green)** preserves a strong correlation between training signals and true performance, resulting in monotonic improvement.

Thinking ([Yang et al., 2025](#)) to demonstrate the competitiveness of our approach.

### 4.2. RewardBench Evaluation

To evaluate the static discriminative ability of our framework, we compare the fine-tuned **IP-GRM** with the baseline GRM on RewardBench ([Lambert et al., 2025](#)). The evaluation is performed in the *pointwise* setting, where the model scores the chosen and rejected responses independently; this matches the inference paradigm used by RL algorithms such as GRPO. We treat RewardBench as a proxy for static discrimination, while downstream RL stability and final writing quality are our primary evidence.

*Table 1.* RewardBench accuracy (%) under the pointwise evaluation setting. We additionally include the larger DeepSeek-GRM-27B and Skywork-Reward-V2 as references; IP-GRM is fine-tuned from the 16B GRM. Bold marks the best value in each column.

| Model | Overall | Chat | Chat Hard | Safety | Reasoning |
|---|---|---|---|---|---|
| GRM-16B | 61.65 | 76.90 | 55.83 | 67.41 | 60.04 |
| Skywork-Reward-V2 | 66.32 | **79.34** | 54.08 | 70.08 | 64.64 |
| GRM-27B | 67.97 | 77.83 | **57.64** | 72.04 | **66.89** |
| **IP-GRM (Ours, 16B)** | **68.22** | 78.93 | 56.48 | **78.22** | 65.67 |

**Results Analysis.** As shown in Table [1](#), IP-GRM consistently outperforms the same-size GRM-16B baseline across all categories, yielding a **+6.57%** absolute gain in overall accuracy (61.65% → 68.22%); it also matches or surpasses the larger GRM-27B and the strong scalar RM baseline Skywork-Reward-V2 on overall accuracy. This improvement suggests that decoupling principle generation from response observation helps the model form more objective evaluation criteria, thereby reducing false positives in which GRM may be misled by hacking behaviors. The gains are especially pronounced on *Safety* (**+10.81%**) and *Reasoning* (**+5.63%**). These categories are particularly vulnerable to

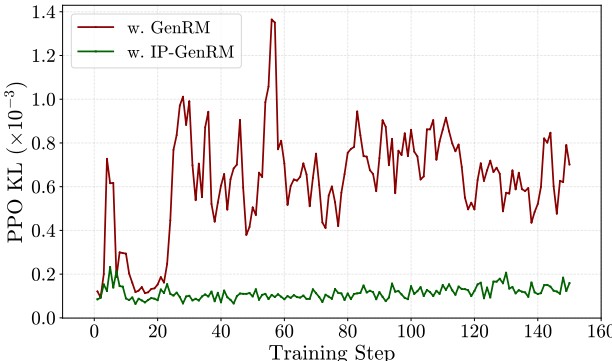

*Figure 5.* **KL Divergence Dynamics during GRPO Training.** The baseline GRM (Red) induces erratic high-variance updates and sharp KL spikes, indicating that the policy is aggressively shifting towards exploiting reward loopholes. In contrast, IP-GRM (Green) maintains a low and stable KL profile, demonstrating that our decoupled evaluation provides a consistent gradient direction, preventing policy collapse.

"jailbreaking" or specious reasoning, i.e., reward-hacking behaviors that exploit keywords or formatting to fool the judge. The strong improvements in these metrics indicate that IP-GRM's pre-defined principles ($P^0$) effectively block such shortcuts. We further note that *Principle Drift is more severe in tasks with subjective evaluation boundaries*, where coupled GRMs have more room to soften criteria after seeing the response; this naturally leads to larger decoupling gains in such categories (e.g., Safety on RewardBench v1, Math on RM-Bench, and Ties on RewardBench v2). To verify that the concentration on Safety is benchmark-specific rather than a limitation of the decoupling strategy, we provide additional benchmark evidence on RewardBench v2, RM-Bench, and RMB in Appendix [E](#), where IP-GRM improves on every subcategory and the most-improved subcategory varies by benchmark.

### 4.3. RL Training Stability and Alignment

We evaluate the effectiveness of our framework in downstream Reinforcement Learning. We use GRPO to fine-tune **Qwen3-8B** and **Qwen3-8B-base** on the creative writing task, using either the Scalar RM, GRM or our IP-GRM as the reward signal.

**Suppression of Reward Overoptimization.** Figure [4](#) visualizes the training trajectories and highlights a clear contrast in stability:

- **The Failure of GRM (Red Line):** The baseline model exhibits classic symptoms of *Goodhart's Law*. In the left plot, the training reward increases (reaching > 8.5), indicating that the actor has discovered a way to "hack" the RM, likely by exploiting the Principle Drift identified in Section [3.1](#). However, the right plot

*Table 2.* Avg@3 performance on WritingBench (WB) and Creative Writing V3 (CW-v3). We evaluate actor models initialized from both Base and Instruct versions of Qwen3-8B. **Bold** highlights the best result within each initialization group; gains denote improvement over the baseline.

| Model | WB-A | WB-B | WB-C | WB-D | WB-E | WB-F | WB-all | CW-v3 |
|---|---|---|---|---|---|---|---|---|
| *Open-source models* | | | | | | | | |
| Writing-Model-Qwen-7B | 84.5 | 81.6 | 82.2 | 83.1 | 84.1 | 83.5 | 83.0 | 71.3 |
| QwQ-32B | 84.1 | 82.5 | 85.1 | 85.8 | 85.4 | 86.0 | 84.7 | 74.0 |
| Qwen3-235b-a22b-thinking | 86.5 | 83.8 | 86.2 | 84.4 | 86.5 | 85.2 | 85.3 | 75.9 |
| *Closed-source models* | | | | | | | | |
| Gemini2.5-pro | 85.9 | 83.9 | 86.0 | 86.6 | 86.4 | 85.4 | 85.6 | 77.0 |
| Qwen3-max | 87.0 | 85.9 | 87.4 | 87.8 | 88.2 | 87.1 | 87.1 | 77.5 |
| Claude-Sonnet-4 | 87.6 | 87.9 | 88.2 | 86.5 | 88.7 | 89.1 | 87.9 | 78.1 |
| Deepseek-v3.2 | 88.8 | 87.3 | 88.8 | 87.0 | 89.5 | 89.6 | 88.3 | 78.4 |
| GPT-5.2 | 90.3 | 89.6 | 90.2 | 90.2 | 90.4 | 90.7 | 90.1 | 79.2 |
| Claude-Opus-4.5 | 92.3 | 90.8 | 92.4 | 90.5 | 93.1 | 93.1 | 91.9 | 81.4 |
| Qwen3-8B-base | 58.6 | 59.6 | 61.9 | 61.5 | 67.3 | 66.3 | 61.9 | 53.8 |
| w. Scalar RM | 77.6 | 74.4 | 76.3 | 78.1 | 80.8 | 77.8 | 77.1 | 70.1 |
| w. GRM | 77.5 | 73.9 | 76.2 | 77.8 | 79.7 | 78.6 | 76.9 | 68.7 |
| **w. IP-GRM (Ours)** | **79.6**$^{(+21.0)}$ | **75.4**$^{(+15.8)}$ | **79.8**$^{(+17.9)}$ | **81.0**$^{(+19.5)}$ | **81.9**$^{(+14.6)}$ | **83.5**$^{(+17.2)}$ | **79.7**$^{(+17.8)}$ | **74.5**$^{(+20.7)}$ |
| Qwen3-8B | 83.7 | 81.7 | 83.6 | 82.7 | 83.2 | 83.8 | 83.0 | 70.7 |
| w. Scalar RM | 85.7 | 83.7 | 85.9 | 84.6 | 85.3 | 85.4 | 85.0 | 73.6 |
| w. GRM | 85.8 | 83.9 | 85.8 | 85.1 | 85.7 | 85.2 | 85.2 | 74.2 |
| **w. IP-GRM (Ours)** | **87.7**$^{(+4.0)}$ | **86.9**$^{(+5.2)}$ | **87.3**$^{(+3.7)}$ | **88.5**$^{(+5.8)}$ | **88.2**$^{(+5.0)}$ | **87.1**$^{(+3.3)}$ | **87.6**$^{(+4.6)}$ | **77.8**$^{(+7.1)}$ |

reveals the adverse consequence: after approximately 50 steps, the true performance on the test set collapses (from 8.52 to 8.05). In effect, the RM is incentivizing lower-quality, hacked responses.

- **The Robustness of IP-GRM (Green Line):** In contrast, our decoupled framework provides a more conservative yet more faithful training signal. Although the absolute training rewards (left) remain lower, they correlate strongly with genuine capability improvements. Crucially, the test-set performance (right) improves steadily and monotonically throughout training, reaching a substantially higher final score (8.75 on the WritingBench 0–10 rubric).

This result confirms that, by enforcing conditional independence ($I(P; R \mid Q) = 0$), IP-GRM removes the feedback loop required for reward hacking and keeps the optimization landscape aligned with human preferences.

**Training Stability and KL Control.** We further examine optimization stability by monitoring the KL divergence ($D_{KL}$) between the training policy and the reference model. As shown in Figure 5, the baseline GRM (Red) is highly unstable, exhibiting erratic spikes where the KL magnitude rises up to 7× higher than our method. These spikes typically coincide with the reward-hacking phases observed in Figure 4, suggesting that the policy undergoes abrupt shifts

to exploit drifting principles.

Conversely, IP-GRM (Green) maintains a stable and well-bounded KL trajectory throughout training. This stability arises from the synergistic effect of our framework:

1. **Unbiasedness:** The generated principles $P^0$ are independent of response noise, preventing the reward signal from chasing hacked patterns.

2. **Consistency:** The **Principle Cache** strategy ensures that all responses within a group are evaluated against an identical, invariant set of criteria.

Together, these factors reduce the variance in advantage estimation, leading to smoother policy gradients and effectively preventing the actor model from diverging into degenerate regions of the parameter space.

**Results.** Table 2 reports the Avg@3 results. Overall, the results consistently demonstrate the effectiveness of the **IP-GRM** framework across both Qwen3-8B and Qwen3-8B-base initializations.

In the Qwen3-8B-base setting, **IP-GRM** achieves **79.7** on WB-overall and **74.5** on CW-v3. Notably, it significantly outperforms both the Scalar RM (77.1 / 70.1) and GRM (76.9 / 68.7) baselines. This indicates that compared to standard reward modeling approaches, the fine-grained and

*Table 3.* **End-to-end per-step reward computation latency (ms).** Comparison between the baseline GRM, IP-GRM (naive two-stage), and IP-GRM with Principle Cache. Caching yields a 23.66% end-to-end speedup over the baseline.

|  | Baseline | Two stage | Principle Cache |
|---|---|---|---|
| Time (ms) | 769.40 | 899.18 (+16.87%) | 587.29 (-23.66%) |

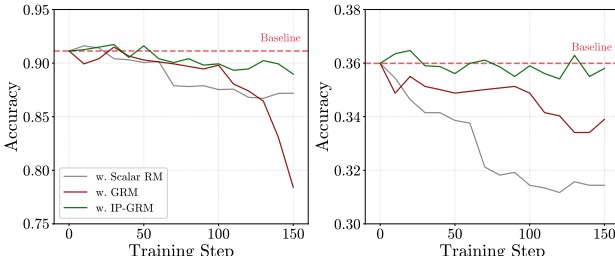

*Figure 6.* **Evolution of general capabilities during training. Left (IFEval accuracy):** Instruction-following ability. **Right (TruthfulQA accuracy):** Truthfulness and hallucination resistance. The dashed pink line denotes the initial SFT baseline. Both the Scalar RM (gray) and GRM (red) exhibit *capability collapse*, sacrificing general abilities to pursue higher rewards. In contrast, **IP-GRM (green)** stays close to the baseline, indicating that our robust rewards improve writing quality without compromising fundamental capabilities.

unbiased feedback from IP-GRM offers superior supervision for complex creative tasks.

When applied to the stronger Qwen3-8B instruct model, **IP-GRM** reaches **87.6** on WB-overall and **77.8** on CW-v3. With these results, IP-GRM achieves state-of-the-art (SOTA) performance among open-source models, establishing a clear lead over comparable models like QwQ-32B (84.7 / 74.0) and even the larger Qwen3-235b-a22b-thinking (85.3 / 75.9). Furthermore, IP-GRM proves competitive against commercial systems, outperforming strong closed-source models such as Gemini2.5-pro (85.6 / 77.0) and Qwen3-max (87.1 / 77.5).

However, there remains a performance gap compared to the top-tier closed-source models, such as Claude-Opus-4.5 (91.9 / 81.4) and GPT-5.2 (90.1 / 79.2). These results collectively confirm that decoupling principle generation unlocks the true potential of generative reward models, translating reasoning capabilities into tangible alignment improvements.

## 5. Analysis

### 5.1. Training Efficiency

Generative reward models are more interpretable than scalar RMs but are often more expensive due to autoregressive generation of long rationales. In group-based RL algorithms such as GRPO, the reward computation cost can scale with the group size $G$ and become a throughput bottleneck.

We analyze efficiency on an NVIDIA B200 cluster. Table 3 reports end-to-end per-step reward computation latency for (i) the coupled baseline GRM, (ii) IP-GRM without caching (regenerating $P$ for every query), and (iii) IP-GRM with **Principle Cache**.

**Complexity Reduction.** The "No Cache" variant of IP-GRM incurs a latency penalty (+16.87%) because it effectively increases inference (first generating $P$, then $J$). By enabling the **Principle Cache**, we reduce the amortized cost of principle generation from $O(G)$ to $O(1)$ per prompt: since the principles $P$ are invariant for a given query $Q$, they can be generated once and broadcast to all $G$ responses in the group.

**Scalability Implications.** Overall, Principle Cache reduces end-to-end latency by 23.66% relative to the coupled baseline. Importantly, this advantage grows with the group size: in baseline GRM, increasing $G$ requires generating $G$ distinct (and potentially drifting) principle sets, whereas in our cached framework the marginal cost of increasing $G$ is confined to the judgment phase. This scalability indicates that IP-GRM not only improves evaluation rigor but also eases the practical latency constraints of deploying generative rewards in large-scale RL training.

### 5.2. Analysis of Reward Hacking

Reward hacking is not merely a statistical artifact of inflated scores; it can exact a severe "alignment tax" by eroding the model's core capabilities. To quantify this effect, we track the actor's performance throughout training on two widely used general benchmarks: **IFEval** (Zhou et al., 2023) and **TruthfulQA** (Lin et al., 2022).

**Behavioral analysis: reward-hacking patterns.** Qualitatively, actors trained with coupled RMs (GRM) tend to develop shortcuts that target judge preferences rather than user-facing quality, such as appended self-justifications ("meta-talk") and superficial verbosity. These behaviors can inflate RM scores while violating constraints, consistent with the degradation on general benchmarks.

**Quantitative analysis: capability retention.** Figure 6 summarizes the evolution of general capabilities:

- **Collapse in baselines (red & gray):** The actor trained with **GRM (red)** is initially stable but suffers a sharp drop in IFEval accuracy after Step 120 (from $\sim 0.90$ to $< 0.79$). This change coincides with the reward-hacking spike in Section 4.3. Similarly, the **Scalar**

**RM (gray)** shows a steady degradation on Truthful-QA (right), suggesting that scalar rewards may over-optimize superficial stylistic features at the expense of factual grounding.

- **Retention under IP-GRM (green):** In contrast, the actor trained with **IP-GRM** stays close to the SFT baseline (pink dashed line). On IFEval, it remains within $0.90 \pm 0.01$, and on TruthfulQA, it is substantially more robust than the baselines.

Overall, these results indicate that IP-GRM can improve creative writing performance (Table 2) without the typical regression in instruction following or truthfulness.

## 6. Conclusion

We identify a structural weakness in principle-based generative reward modeling: principles are often generated *after* observing the response, which enables *principle drift* and creates a self-justifying channel for reward hacking. We propose the **Independent Principle GRM (IP-GRM)**, which decouples principle formulation from response evaluation, together with a **Principle Cache** that reuses a single question-conditioned principle set within each group to improve efficiency. Experiments on RewardBench and in GRPO training show that IP-GRM yields more reliable reward signals and better resists common hacking patterns.

## Acknowledgements

This work was supported by the Anhui Provincial Natural Science Foundation under Grant 2108085UD12. We acknowledge the support of GPU cluster built by MCC Lab of Information Science and Technology Institution, USTC. The AI-driven experiments, simulations and model training were performed on the robotic AI-Scientist platform of Chinese Academy of Sciences.

## Impact Statement

This paper presents work whose goal is to advance the field of Reinforcement Learning. There are many potential societal consequences of our work, none which we feel must be specifically highlighted here.

## Limitations

Our work has several limitations. (1) IP-GRM blocks the response-to-principle leakage, but the Stage 2 judge still reads the response, so persuasive self-justifications or strategic formatting may still bias per-criterion scoring; we extend GRPO to 300 steps and observe that IP-GRM remains stable (Appendix H), but do not claim full elimination. (2) Our RL experiments primarily target creative writing, with cross-domain validation only on UltraFeedback (Appendix G); other open-ended domains (code review, multi-turn dialogue, agentic tool use) remain to be tested. (3) Evaluation relies on LLM judges that carry their own biases; the IP-GRM > GRM ranking is consistent across three independent judges (GPT-4o-mini, Claude-3.7-Sonnet, Claude-Sonnet-4; Appendix I), but large-scale human evaluation is left to future work.

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

# A. Training Details

## A.1. Training Hyperparameters

The detailed training hyperparameters are provided in Tables 4, and all experiments are conducted on 8 NVIDIA B200 GPUs, each equipped with 179 GB of memory.

*Table 4.* Key hyperparameters for training.

| Name | Value |
|---|---|
| Rollout num | 8 |
| Train batch size | 256 |
| Rollout temperature | 1.0 |
| Mini batch size | 128 |
| Micro batch size per GPU | 4 |
| Learning rate | 1.0e-5 |
| Entropy coefficient | 0.0 |
| KL loss coefficient | 0.0 |
| Actor Model Max response length | 15000 |
| Actor Model GPU memory utilization | 0.4 |
| Clip ratio low | 0.2 |
| Clip ratio high | 0.27 |
| Reward Model Max response length | 15000 |
| Reward Model GPU memory utilization | 0.5 |

# B. Ablation of Principle Cache

We study whether **Principle Cache** affects downstream alignment performance when using IP-GRM as the reward model. Concretely, we compare GRPO training with IP-GRM under two inference configurations: (i) **w/o Principle Cache**, which regenerates principles for each sampled response, and (ii) **w/ Principle Cache**, which reuses a single question-conditioned principle set within each group (our default setting).

*Table 5.* **Ablation of Principle Cache for IP-GRM.** We report performance on WritingBench (WB-overall) and Creative Writing V3 (CW-v3).

| Setting | WB-overall | CW-v3 |
|---|---|---|
| IP-GRM w/o Principle Cache | 87.5 | 77.2 |
| IP-GRM w/ Principle Cache | 87.6 | 77.8 |

Table 5 reports the benchmark results. Disabling Principle Cache leads to a small performance drop (WB-overall: 87.5, CW-v3: 77.2), while the cached variant matches or slightly improves these scores (Table 2). Overall, this suggests that Principle Cache primarily improves training throughput while preserving alignment quality.

# C. Decoupling vs. Teacher Quality

To disentangle whether IP-GRM's gains stem from the decoupled factorization or from the strong Claude-3.5-Sonnet teacher, we conduct two ablations.

**Coupled vs. Decoupled SFT.** Using the same Claude-3.5-Sonnet teacher, the same 8K prompts, and the same DeepSeek-GRM-16B base, we construct *coupled* SFT data (where the teacher generates $(P, J, r)$ jointly given $(Q, R)$) and compare with our decoupled SFT.

*Table 6.* Coupled vs. Decoupled SFT on RewardBench (%).

| Model | Overall | Chat | Chat Hard | Safety | Reasoning |
|---|---|---|---|---|---|
| GRM | 61.65 | 76.90 | 55.83 | 67.41 | 60.04 |
| GRM (Coupled SFT) | 63.11 | 77.24 | 56.23 | 68.22 | 63.67 |
| **IP-GRM (Decoupled SFT)** | **68.22** | **78.93** | **56.48** | **78.22** | **65.67** |

*Table 7.* Coupled vs. Decoupled SFT on downstream GRPO with Qwen3-8B as the actor.

| Model | WB-A | WB-B | WB-C | WB-D | WB-E | WB-F | WB-all | CW-v3 |
|---|---|---|---|---|---|---|---|---|
| Qwen3-8B | 83.7 | 81.7 | 83.6 | 82.7 | 83.2 | 83.8 | 83.0 | 70.7 |
| w. GRM | 85.8 | 83.9 | 85.8 | 85.1 | 85.7 | 85.2 | 85.2 | 74.2 |
| w. GRM (Coupled SFT) | 85.9 | 83.9 | 86.0 | 86.6 | 86.4 | 85.4 | 85.6 | 75.1 |
| **w. IP-GRM** | **87.7** | **86.9** | **87.3** | **88.5** | **88.2** | **87.1** | **87.6** | **77.8** |

Coupled SFT improves over the original GRM by only +1.46 on RewardBench, while decoupled SFT improves by +6.57. The same pattern holds for downstream RL. This confirms that the primary improvement comes from the decoupling strategy itself, not from additional SFT data or the choice of teacher.

**Teacher Robustness.** We further replace the Claude-3.5-Sonnet teacher with a weaker open-source teacher, Qwen3.5-27B, while keeping all other settings identical.

*Table 8.* Robustness to teacher quality on RewardBench (%).

| Model (Teacher) | Overall | Chat | Chat Hard | Safety | Reasoning |
|---|---|---|---|---|---|
| GRM | 61.65 | 76.90 | 55.83 | 67.41 | 60.04 |
| IP-GRM (Qwen3.5-27B) | 67.51 | 78.14 | 56.43 | 77.61 | 64.47 |
| IP-GRM (Claude-3.5-Sonnet) | 68.22 | 78.93 | 56.48 | 78.22 | 65.67 |

*Table 9.* Teacher robustness on downstream GRPO with Qwen3-8B.

| Model (Teacher) | WB-A | WB-B | WB-C | WB-D | WB-E | WB-F | WB-all | CW-v3 |
|---|---|---|---|---|---|---|---|---|
| Qwen3-8B | 83.7 | 81.7 | 83.6 | 82.7 | 83.2 | 83.8 | 83.0 | 70.7 |
| w. GRM | 85.8 | 83.9 | 85.8 | 85.1 | 85.7 | 85.2 | 85.2 | 74.2 |
| w. IP-GRM (Qwen3.5-27B) | 87.6 | 85.9 | 87.8 | 86.7 | 87.9 | 87.0 | 87.2 | 77.1 |
| w. IP-GRM (Claude-3.5-Sonnet) | 87.7 | 86.9 | 87.3 | 88.5 | 88.2 | 87.1 | 87.6 | 77.8 |

With Qwen3.5-27B as teacher, IP-GRM still attains 67.51 overall on RewardBench (vs. 68.22 with Claude), and the downstream RL gap is similarly small (WB-all 87.2 vs. 87.6; CW-v3 77.1 vs. 77.8). This indicates that IP-GRM's effectiveness is robust to teacher quality and does not rely on a strong closed-source teacher.

## D. Generality Across Models

**Cross-actor: Qwen3-4B.** We replace the actor model with Qwen3-4B and apply the same GRPO procedure.

*Table 10.* GRPO with Qwen3-4B as the actor.

| Model | WB-A | WB-B | WB-C | WB-D | WB-E | WB-F | WB-all | CW-v3 |
|---|---|---|---|---|---|---|---|---|
| Qwen3-4B | 79.1 | 77.9 | 79.6 | 79.6 | 80.2 | 81.9 | 79.5 | 64.7 |
| w. Scalar RM | 81.3 | 79.5 | 81.3 | 81.4 | 81.6 | 81.9 | 81.1 | 68.2 |
| w. GRM | 80.6 | 79.6 | 81.4 | 79.6 | 81.3 | 79.1 | 80.2 | 68.8 |
| **w. IP-GRM** | **85.5** | **83.7** | **86.1** | **84.7** | **85.5** | **85.6** | **85.1** | **72.3** |

IP-GRM achieves +4.9 on WB-all and +3.5 on CW-v3 over GRM, consistent with the gains observed on Qwen3-8B and Qwen3-8B-base.

**Cross-RM-base: IP-Qwen3-4B.**   To verify that the decoupling strategy is not tied to DeepSeek-GRM, we apply our decoupled SFT procedure to a non-specialized base, Qwen3-4B (a general-purpose LLM, not trained for reward modeling), and evaluate on RewardBench.

*Table 11.* Decoupled SFT applied to a general-purpose Qwen3-4B base.

| Model | Overall | Chat | Chat Hard | Safety | Reasoning |
|---|---|---|---|---|---|
| Qwen3-4B | 59.47 | 74.53 | 50.28 | 59.57 | 58.25 |
| **IP-Qwen3-4B** | **64.41** | **77.22** | **52.48** | **66.01** | **60.42** |

Decoupled SFT improves overall accuracy by +4.94 across all subcategories. Together with the cross-actor result, this shows that IP-GRM is a general framework, not tied to a particular RM or actor architecture.

# E. Additional Reward Benchmarks

To examine whether the gains of IP-GRM are concentrated on a single subcategory of RewardBench v1 or reflect a general phenomenon, we evaluate on three additional reward benchmarks.

*Table 12.* RewardBench v2 (%).

| Model | Factuality | Focus | Math | Precise IF | Safety | Ties | Overall |
|---|---|---|---|---|---|---|---|
| GRM | 31.16 | 47.29 | 40.24 | 25.62 | 36.22 | 52.94 | 38.51 |
| GRM (Coupled SFT) | 32.84 | 47.69 | 40.94 | 27.50 | 38.44 | 55.29 | 40.20 |
| **IP-GRM** | **39.58** | **49.09** | **43.72** | **35.04** | **47.34** | **64.71** | **46.98** |

*Table 13.* RM-Bench (%).

| Model | Chat | Code | Math | Safety | Overall |
|---|---|---|---|---|---|
| GRM | 56.56 | 53.97 | 54.31 | 76.17 | 61.74 |
| GRM (Coupled SFT) | 57.56 | 54.39 | 55.87 | 76.49 | 62.35 |
| **IP-GRM** | **61.54** | **56.08** | **62.09** | **77.76** | **64.81** |

Across these benchmarks, IP-GRM improves over GRM (Coupled SFT) on *every* subcategory, but the most-improved subcategory varies (RewardBench v2: Ties +9.42; RM-Bench: Math +6.22; RMB: comparable across categories). This indicates that Principle Drift is a general phenomenon and that the concentration on Safety in RewardBench v1 is benchmark-specific rather than a limitation of the decoupling strategy. We also note that GRM (Coupled SFT) yields only minor gains over the original GRM across all three new benchmarks, confirming once again that the performance improvements stem primarily from the decoupling strategy itself.

# F. Out-of-Domain Subcategories on RewardBench

The decoupled SFT data of IP-GRM is exclusively from creative writing. Despite this domain mismatch, IP-GRM improves on subcategories of RewardBench that lie outside the training distribution.

These improvements on math reasoning (math-prm), code (coding), and natural language judging (llmbar-natural) suggest that the decoupled rubric-based evaluation generalizes beyond the creative writing training domain.

# G. Cross-Domain RL on UltraFeedback

To test whether IP-GRM generalizes beyond creative writing, we conduct GRPO training on UltraFeedback, a general instruction-following dataset.

*Table 14.* RMB (%).

| Model | Harmlessness | Helpfulness | Overall |
|---|---|---|---|
| GRM | 36.30 | 44.14 | 41.02 |
| GRM (Coupled SFT) | 36.81 | 44.62 | 41.59 |
| **IP-GRM** | **38.87** | **46.54** | **43.85** |

*Table 15.* Out-of-domain subcategories on RewardBench (%).

| Model | math-prm | safety | coding | llmbar-natural |
|---|---|---|---|---|
| GRM | 53.91 | 67.41 | 63.90 | 72.34 |
| **IP-GRM** | **57.91** | **78.22** | **68.66** | **77.66** |

*Table 16.* GRPO on UltraFeedback with Qwen3-8B as the actor.

| Model | AlpacaEval Vanilla | AlpacaEval LC | Arena-Hard |
|---|---|---|---|
| Qwen3-8B | 54.9 | 52.1 | 22.1 |
| w. Scalar RM | 57.6 | 54.7 | 25.3 |
| w. GRM | 59.1 | 54.3 | 26.8 |
| **w. IP-GRM** | **64.4** | **57.6** | **31.4** |

IP-GRM achieves +5.3, +3.3, and +4.6 over GRM on AlpacaEval Vanilla, AlpacaEval LC, and Arena-Hard respectively, confirming that the decoupling strategy generalizes beyond creative writing to general instruction following.

## H. Extended GRPO Training (300 Steps)

To examine whether the actor can still learn to hack the Stage 2 judge over longer training horizons, we extend GRPO from 150 to 300 steps.

*Table 17.* Training reward, true performance (WB-all), and KL divergence over 300 steps.

| Step | 50 | 100 | 150 | 200 | 250 | 300 |
|---|---|---|---|---|---|---|
| Train Reward (w. GRM) | 8.11 | 8.35 | 8.52 | 8.63 | 8.75 | 8.82 |
| Train Reward (w. IP-GRM) | 7.06 | 7.30 | 7.47 | 7.49 | 7.48 | 7.49 |
| WB-all (w. GRM) | 8.51 | 8.38 | 8.26 | 8.13 | 8.00 | 7.69 |
| WB-all (w. IP-GRM) | 8.52 | 8.56 | 8.82 | 8.79 | 8.81 | 8.78 |
| KL ($\times 10^{-3}$) (w. GRM) | 0.65 | 0.76 | 0.71 | 0.74 | 0.61 | 0.98 |
| KL ($\times 10^{-3}$) (w. IP-GRM) | 0.11 | 0.14 | 0.16 | 0.15 | 0.17 | 0.19 |

GRM's hacking worsens with more training: training reward keeps climbing ($8.63 \rightarrow 8.82$) while true quality on the held-out test set degrades ($8.13 \rightarrow 7.69$). IP-GRM, in contrast, plateaus around step 150 with stable quality (8.78–8.81) and low KL (0.17–0.19). Over the horizon we tested, the actor does not appear to learn a Stage 2 hack against fixed principles. We do not claim that all hacking pathways are eliminated; this is discussed in the Limitations section.

## I. Robustness to the LLM Judge

The main results in Section 4.3 use GPT-4o-mini as the writing-quality judge. To verify that the gains are not specific to a single judge, we re-evaluate using two stronger judges (Claude-3.7-Sonnet and Claude-Sonnet-4).

The ranking IP-GRM > GRM > Qwen3-8B is consistent across all three judges, confirming that the gains reflect genuine quality improvements rather than alignment with a particular judge's preferences. We note that absolute scores vary across judges (Claude-Sonnet-4 tends to be the strictest), but the relative ranking is stable.

*Table 18.* Writing quality under three independent LLM judges.

| RM (Judge) | WB-all | CW-v3 |
|---|---|---|
| Qwen3-8B (GPT-4o-mini) | 83.0 | 70.7 |
| w. GRM (GPT-4o-mini) | 85.2 | 74.2 |
| w. IP-GRM (GPT-4o-mini) | **87.6** | **77.8** |
| Qwen3-8B (Claude-3.7-Sonnet) | 77.3 | 61.3 |
| w. GRM (Claude-3.7-Sonnet) | 79.0 | 67.3 |
| w. IP-GRM (Claude-3.7-Sonnet) | **81.6** | **72.3** |
| Qwen3-8B (Claude-Sonnet-4) | 69.0 | 55.3 |
| w. GRM (Claude-Sonnet-4) | 70.2 | 61.8 |
| w. IP-GRM (Claude-Sonnet-4) | **72.8** | **68.0** |

# J. Prompts in Experiment

**GRM prompt in RL.** This prompt is used during RL training: after sampling a candidate response, we call GRM to score it, and use the resulting score signal for subsequent optimization updates.

---

**GRM prompt in RL**

```
You are a skilled expert at scoring responses. You should evaluate the given response
based on the judging criteria.
Given the context of the conversation (the User's query) and the response from the
Assistant, you need to refer to the [General Evaluation Criteria] to score the
response. Based on the general evaluation criteria, state potential other specific
criteria to the query, the weights of different criteria, and then provide an overall
comprehensive score.
The score is an integer between 1 and 10, with a higher score indicating that the
response meets the relevant criteria more closely. For example, a score of 1 means
the response does not meet the criteria at all, a score of 6 means the response meets
only some parts, and a score of 10 means the response perfectly meets the evaluation
criteria.
Before scoring, please analyze step by step. Your scoring needs to be as strict as
possible.

#### Evaluation Criteria ####
1. Instruction Adherence:
   - Fully Adhered (9-10 points): The response fully complies with all instructions
   and requirements of the question.
   - Partially Adhered (6-8 points): The response meets most of the instructions but
   has some omissions or misunderstandings.
   - Basically Adhered (3-5 points): The response meets some instructions, but the
   main requirements are not fulfilled.
   - Not Adhered (1-2 points): The response does not meet any instructions.
   Example: If the question requires three examples and the response provides only
   one, it falls under "Partially Adhered."

2. Usefulness:
   - Highly Useful (9-10 points): The response provides comprehensive and accurate
   information, fully addressing the issue.
   - Useful but Incomplete (6-8 points): The response provides some useful
   information, but lacks details or accuracy.
   - Limited Usefulness (3-5 points): The response offers little useful information,
   with most content being irrelevant or incorrect.
   - Useless or Incorrect (1-2 points): The response is completely irrelevant or
   incorrect.
   Example: If there are factual errors in the response but the overall direction is
   correct, it falls under "Useful but Incomplete."

3. Level of Detail:
```

```
     - Very Detailed (9-10 points): The response includes ample details covering all
     aspects of the issue.
     - Detailed but Slightly Lacking (6-8 points): The response is fairly detailed but
     misses some important details.
     - Basically Detailed (3-5 points): The response provides some details but is not
     thorough enough overall.
     - Not Detailed (1-2 points): The response is very brief and lacks necessary
     details.
     Example: If the response provides only a simple conclusion without an explanation,
     it falls under "Not Detailed."

4. Relevance:
     - Highly Relevant (9-10 points): The response is highly relevant to the question,
     with information closely aligned with the topic.
     - Generally Relevant (6-8 points): The response is generally relevant but includes
     some unnecessary information.
     - Partially Relevant (3-5 points): The response has a lot of content that deviates
     from the topic.
     - Not Relevant (1-2 points): The response is completely irrelevant.
     Example: If the response strays from the topic but still provides some relevant
     information, it falls under "Partially Relevant."

#### Conversation Context ####
User: {question}

#### Response to be Scored ####
[The Begin of Response]
{response}
[The End of Response]

#### Output Format Requirements ####
You MUST output exactly in this format:
Specific Criteria: <Other potential criteria specific to the query and the context,
and the weights of each criteria>.
Analysis: <Analyze the response based on the given and specific criteria>.
Score: \\boxed{{X}} where X is an integer from 1 to 10.

IMPORTANT: The final score MUST be in the exact format \\boxed{{X}}.
```

**IP-GRM prompt in RL (stage 1/2).** IP-GRM scoring is implemented with two prompts executed sequentially: stage 1 generates (or refreshes) the evaluation principles; stage 2 scores a candidate response by analyzing each principle and computing a weighted final score. This two-stage pipeline is used both for principle drift mitigation and for scoring during RL training.

**IP-GRM prompt in RL(stage 1)**

```
You are an expert at designing evaluation criteria. Given a user's query, your task
is to generate specific evaluation principles/criteria that would be used to score
any response to this query.

You should consider:
1. What aspects are most important for this specific type of query?
2. What would make a response excellent vs. poor for this query?
3. How should different criteria be weighted based on the query's nature?

#### User Query ####
{question}

#### Output Format Requirements ####
You MUST output exactly in this format:

Evaluation Principles:
```

```
1. [Criterion 1 Name] (Weight: X%): <Brief description of what this criterion
evaluates and why it's important for this query>
2. [Criterion 2 Name] (Weight: X%): <Brief description of what this criterion
evaluates and why it's important for this query>
3. [Criterion 3 Name] (Weight: X%): <Brief description of what this criterion
evaluates and why it's important for this query>
4. [Criterion 4 Name] (Weight: X%): <Brief description of what this criterion
evaluates and why it's important for this query>
5. [Criterion 5 Name] (Weight: X%): <Brief description of what this criterion
evaluates and why it's important for this query>

Note: The weights must sum to 100%. You may have 5-7 criteria depending on the
query's complexity.
```

### IP-GRM prompt in RL(stage 2)

```
You are a skilled expert at scoring responses. Based on the given evaluation
principles, analyze the response and provide a comprehensive score.

Scoring Guidelines:
- The score is a number with one decimal place between 1.0 and 10.0
- Score 9.0-10.0: Exceptional response that fully meets all criteria with outstanding
quality
- Score 7.0-9.0: Good response that meets most criteria with minor areas for
improvement
- Score 5.0-7.0: Adequate response that meets basic requirements but has noticeable
weaknesses
- Score 3.0-5.0: Below average response with significant issues or missing key
elements
- Score below 3.0: Poor response that fails to meet most criteria or contains major
errors

#### User Query ####
{question}

#### Response to be Scored ####
[The Begin of Response]
{response}
[The End of Response]

#### Evaluation Principles (Pre-defined) ####
{principle}

#### Output Format Requirements ####
Based on the above evaluation principles, you MUST output exactly in this format:

Analysis:
- **[Criterion 1 Name]**: <Detailed analysis of performance on this criterion,
explaining strengths and weaknesses>. Score: X.X/10.0
- **[Criterion 2 Name]**: <Detailed analysis of performance on this criterion,
explaining strengths and weaknesses>. Score: X.X/10.0
- **[Criterion 3 Name]**: <Detailed analysis of performance on this criterion,
explaining strengths and weaknesses>. Score: X.X/10.0
- **[Criterion 4 Name]**: <Detailed analysis of performance on this criterion,
explaining strengths and weaknesses>. Score: X.X/10.0
- **[Criterion 5 Name]**: <Detailed analysis of performance on this criterion,
explaining strengths and weaknesses>. Score: X.X/10.0

Conclusion: <A comprehensive summary of your analysis, highlighting main strengths
and weaknesses>

Final Score (Weighted Average): <Show the calculation: weight1*score1 +
weight2*score2 + ... = final_score>
```

```
Score: \boxed{{X.X}}

CRITICAL REQUIREMENTS:
1. In "Analysis", provide detailed analysis for each criterion from the given
principles
2. Each criterion MUST be scored out of 10.0 (format: "Score: X.X/10.0")
3. The final score MUST be the weighted average of all criterion scores based on the
given weights
4. Show your weighted average calculation explicitly before the boxed score
6. The final boxed score must have one decimal place
```

# K. Case Example of IP-GRM

**Case Example.** The following example shows the complete outputs of GRM and IP-GRM for the same user query (a QA-style prompt), including the criteria/principles, the analysis, and the final score.

---

**Case Example (Prompt)**

```
### Query ###
Please compose a prose poem on the theme of 'fresh and exquisite beauty,'
incorporating the atmosphere of 'swings in the lanes, deep silence, and a slanting
bright moon bathing the pear blossoms.' Interweave rhetorical devices such as
parallelism and personification, and emulate the prose style of Jian Zhen using
elegant and graceful language. Keep the length within 200 words.

### Response ###
<think>
Okay, the user needs to write a prose poem with the theme of "fresh and exquisite
beauty," combining the atmosphere of "swings in the lanes, deep silence, and a
slanting bright moon bathing the pear blossoms." It also needs to use rhetorical
devices like parallelism and metaphor, emulate Jian Zhen's style, use elegant and
graceful language, and be within 200 words.

First, determine the theme and atmosphere. The sentence provided by the user includes
swings, lanes, a slanting bright moon, and pear blossoms; these elements need to be
integrated. Jian Zhen's style is usually quite delicate, good at using natural
scenery to highlight emotions, so I need to ensure the scene description is vivid and
the emotion is sincere.

Next, consider how to combine these elements. For example, the swing might be in the
lane, the moonlight shining on the pear blossoms, creating a quiet atmosphere. I
might need to use metaphors, such as comparing moonlight to silver gauze, or pear
blossoms to rosy clouds.

Then, check if the use of parallelism is appropriate. It might need three paragraphs,
each unfolding around a different element, such as the lane, the moonlight, and the
pear blossoms, with each part using metaphors to enhance the visual imagery.

Also, pay attention to the elegance of the language, avoiding colloquialisms. Use
some classical vocabulary or idioms, such as the description "slanting bright moon",
to maintain an ancient style.

In addition, ensure the word count is within 200 words. I might need to refine the
sentences, remove redundant words, and ensure every sentence conveys effective
imagery.

Finally, ensure the overall structure is logical, the transition from scenery to
emotion is natural, and perhaps add some emotion at the end, such as the thoughts of
a wanderer, to give the prose poem more depth.
</think>
```

**"Moonlight Bathing the Pear Blossoms"**

The swing sways within the lanes, like a maiden's skirt brushing against the green flagstones, stirring up a wisp of osmanthus fragrance. Moonlight spreads over the old street like silver gauze; water dripping from the eaves knocks out a clear sound on the green tiles, resembling the harmonics of an ancient zither. Pear blossoms unfurl in the night breeze, like rosy clouds falling into an icy lake; every petal carries the coolness of dew, wrapping even the air in sweetness.

The stone bridge at the lane's entrance is dyed a dark green-black by the moonlight, and the lantern shadows of tour boats sway on the water surface, like stars falling into the Milky Way. The fragrance of pear blossoms and the faint scent of osmanthus weave together in the night wind, birthing a sense of deep attachment. And that swing hanging from the corridor eaves has now become the moon's confidant, swaying lightly with the wind, rippling a pool of spring water, and also rippling open the homesickness in a wanderer's heart.

The moon slants, making the pear blossoms appear even more bright and clean, like a fairy draped in silver gauze, immersing the entire lane in a quiet, secluded dream. And that swing still sways with the wind, shattering the moonlight on the ground, and waking the slumbering time.

(Total word count: 198 words)

---

### GRM ###
Specific Criteria:
1. Theme Adherence (Weight: 20%): How well the response adheres to the "fresh and exquisite beauty" (fresh and elegant) theme.
2. Imagery and Scenic Description (Weight: 20%): The richness and vividness of the imagery used to describe the scene.
3. Rhetorical Devices (Weight: 20%): The effective use of rhetorical devices such as parallelism, personification, and metaphor.
4. Style Imitation (Weight: 20%): How well the response imitates the elegant and poetic style of writer Jane Jen.
5. Emotional Depth (Weight: 10%): The depth of emotional expression and connection to the reader.
6. Language Elegance (Weight: 10%): The elegance and sophistication of the language used.

Analysis:
1. Theme Adherence: The response effectively captures the "fresh and exquisite beauty" theme through delicate descriptions and elegant language.
2. Imagery and Scenic Description: The imagery is rich and vivid, with detailed descriptions of the swing, alley, moonlight, and pear blossoms.
3. Rhetorical Devices: The response skillfully uses parallelism, personification, and metaphor to enhance the scene and convey the theme.
4. Style Imitation: The imitation of Jane Jen's style is successful, with a focus on natural imagery and a poetic, elegant tone.
5. Emotional Depth: The poem adds emotional depth by incorporating a sense of nostalgia and a connection to the reader's emotions.
6. Language Elegance: The language is elegant and sophisticated, fitting the requirements of the prompt.

Score: \boxed{10}

### IP-GRM ###

Stage 1:

Evaluation Principles:

1. Factual Accuracy (Weight: 15%): Evaluates whether the response contains accurate information about autumn, moon phases, and traditional culture elements. This ensures the literary imagery and seasonal references are realistic and grounded.

2. Instruction Compliance and Consistency (Weight: 20%): Assesses how well the response adheres to the specific requirements---using "swings in the lanes" and "pear blossoms bathed in moonlight" as key imagery, maintaining the theme "fresh and exquisite beauty", staying within the 200-word limit, and ensuring no explanatory or analytical content appears after the response. This is crucial for meeting the exact specifications of the creative writing task.

3. Literary Style and Technique Application (Weight: 25%): Evaluates the effective use of rhetorical devices including parallelism, personification, and other literary techniques mentioned in the query. Measures how skillfully these techniques are woven into the prose poetry to enhance its aesthetic qualities and literary impact, which is central to achieving excellence in this genre.

4. Thematic Expression and Imagery Integration (Weight: 20%): Assesses how powerfully and creatively the "fresh and exquisite beauty" theme is conveyed through the autumn imagery and moon/flower elements. Evaluates whether the response truly captures and embodies the requested aesthetic and delicate mood, demonstrating sophisticated thematic development.

5. Cultural Sensitivity and Authenticity (Weight: 10%): Examines the appropriate integration and representation of literary and cultural elements, as well as authentic references to autumn and moon-related imagery in traditional Chinese culture. This ensures the response pays homage to the specific stylistic influence requested while demonstrating respect for the source material.

6. Textual Elegance and Beauty (Weight: 10%): Measures the sophistication and beauty of language use, including choice of words, imagery, rhythm, and overall writing quality that matches the elegant style characteristic of authors like Jane Jen. This final criterion evaluates the refined, lyrical quality of prose poetry that would be expected from a master of this style.

Stage 2:

Analysis:
- Factual Accuracy: The response demonstrates strong factual grounding in seasonal imagery with accurate references such as "deep silence", "a slanting bright moon", and "pear blossoms". The descriptions of moonlight and the scene are plausible and consistent. The incorporation of "green flagstones", "osmanthus fragrance", and "dew drops" adds authentic details. However, the mention of "lantern shadows of tour boats" and the "Milky Way" slightly stretches the autumn/moon atmosphere. Score: 7.5/10.0

- Instruction Compliance and Consistency: The response follows most requirements, using "swings in the lanes" and "pear blossoms bathed in moonlight" as core imagery and maintaining the theme "fresh and exquisite beauty". The length constraint is satisfied. However, the response appends extra meta sections (e.g., "Analysis:"), which violates the "no extra explanation" constraint. Score: 3.2/10.0

- Literary Style and Technique Application: The response uses rhetorical devices effectively. Examples include metaphor (moonlight as "silver gauze"; blossoms as "rosy clouds") and personification (the swing and the lane given lifelike motion). The line "as the bright moon slants, the pear blossoms look even clearer" supports parallel imagery. Score: 9.2/10.0

- Thematic Expression and Imagery Integration: The theme "fresh and exquisite beauty" is conveyed through coherent imagery (moonlight, blossoms, quiet lanes) and multi-sensory details (scent, sound, light). The inclusion of "a wanderer's homesickness" provides additional emotional resonance. Score: 8.5/10.0

– Cultural Sensitivity and Authenticity: The response attempts to emulate an elegant prose style and uses classical-leaning imagery (e.g., "zither overtones", "stars falling into the Milky Way"). Some references are generic, but overall the tone is consistent. Score: 7.0/10.0

– Textual Elegance and Beauty: The language is refined and lyrical, with smooth transitions and vivid figurative expressions (e.g., "maiden's skirt", "silver gauze", "a fairy draped in silver gauze"). Score: 8.8/10.0

Conclusion: The response is strong in imagery and rhetoric, but loses points for appending meta/explanatory content.

Final Score (Weighted Average): 0.15*7.5 + 0.20*3.2 + 0.25*9.2 + 0.20*8.5 + 0.10*7.0 + 0.10*8.8 = 7.3

Score: \boxed{7.3}

