# OpenReview forum: "Unbiased Principles, Robust Rewards"
_ICML.cc/2026/Conference — ICML 2026 regular_

### Official Review · Reviewer_mTjs · 2026-03-03

**Soundness:** 2
**Presentation:** 2
**Significance:** 2
**Originality:** 2
**Overall Recommendation:** 3
**Confidence:** 4

**Summary:**

The paper studies Generative Reward Models (GRMs) for principle drift. They shows that it is caused because mutual information between evaluation principles and response (R) in standard GRMs is non zero. They further propose Independent Principle GRM (IP-GRM) which decouples evaluation principles (P) and judgement (J) by a two stage inference procedure where P is conditioned only on question (Q) and then J independently sampled from policy (conditioned on P, Q and R). Further, they provide a caching mechanism for efficient implementation and usage of IP-GRM during GRPO training.

**Compliance With Llm Reviewing Policy:**

Affirmed.

**Final Justification:**

The additional experiments provided by the authors address my concerns regarding baseline comparisons. However, I still have reservations about the effectiveness of the method. The gains achieved by the method are inconsistent across datasets, and the authors need a better mechanism to explain this phenomenon rather than simply presenting improved results on different datasets. I am increasing my score from 2 -> 3, as I believe the paper has merit but requires revisions before it can be meaningfully built upon by others.

**Key Questions For Authors:**

Please refer Strengths And Weaknesses section.

**Limitations:**

yes

**Strengths And Weaknesses:**

### Strength
---
- Paper is well written and easy to follow with clear motivation and problem formulation.
- Authors provide detailed ablations and mathematical grounding to their approach.

### Weakness
----
- The paper uses Claude-3.5-Sonnet for creating decoupled SFT dataset (i.e., evaluation principles (P) and judgement (J)) with responses (R) generated from Qwen3-8B. Then this data is used to finetune DeepSeekGRM-16B to produce IP-GRM. It is bit ambiguous that gains in performance are coming because of using a stronger teacher (Claude) to train GRM or from the decoupling strategy itself. I request the authors to finetune base GRM model on same datapoints using coupled SFT data generated from Claude.
- Further, training dataset and response generating model (Qwen3-8B) used for GRM and GRPO downstream training are same which could create leakage from stronger teacher. I would like to see what happens when we use Qwen3-8B or similar open source model as teacher model instead of much powerful Claude model.
- There are inconsistency in abbreviations which are used i.e., GenRM / GRM, IP-GenRM/IP-GRM

---

> ### Author Rebuttal · Authors · 2026-03-29
>
> We sincerely thank the reviewer for the feedback and the recognition of our clear motivation and mathematical grounding. We address each concern below.
>
> > **W1:** It is ambiguous that gains in performance are coming because of using a stronger teacher (Claude) to train GRM or from the decoupling strategy itself. I request the authors to finetune base GRM model on same datapoints using coupled SFT data generated from Claude.
>
> This is a valid concern. We conduct the requested ablation: using the same Claude-3.5-Sonnet teacher, the same 8K prompts, and the same DeepSeek-GRM-16B base model, we construct **coupled** SFT data and compare with our decoupled SFT (IP-GRM).
>
> | Model | Overall | Chat | Chat Hard | Safety | Reasoning |
> |-------|---------|------|-----------|--------|-----------|
> | GRM | 61.65 | 76.90 | 55.83 | 67.41 | 60.04 |
> | GRM (Coupled SFT) | 63.11 | 77.24 | 56.23 | 68.22 | 63.67 |
> | **IP-GRM (Decoupled SFT)** | **68.22** | **78.93** | **56.48** | **78.22** | **65.67** |
>
> Coupled SFT improves over the original GRM by only +1.46, while decoupled SFT improves by +6.57. The same pattern holds in downstream RL:
>
> | Model | WB-A | WB-B | WB-C | WB-D | WB-E | WB-F | WB-all | CW-v3 |
> |-------|------|------|------|------|------|------|--------|-------|
> | Qwen3-8B | 83.7 | 81.7 | 83.6 | 82.7 | 83.2 | 83.8 | 83.0 | 70.7 |
> | w. GRM | 85.8 | 83.9 | 85.8 | 85.1 | 85.7 | 85.2 | 85.2 | 74.2 |
> | w. GRM (Coupled SFT) | 85.9 | 83.9 | 86.0 | 86.6 | 86.4 | 85.4 | 85.6 | 75.1 |
> | **w. IP-GRM (Decoupled SFT)** | **87.7** | **86.9** | **87.3** | **88.5** | **88.2** | **87.1** | **87.6** | **77.8** |
>
> This confirms that the primary improvement stems from the **decoupling strategy**.
>
> > **W2:** Training dataset and response generating model (Qwen3-8B) used for GRM and GRPO downstream training are same which could create leakage from stronger teacher. I would like to see what happens when we use Qwen3-8B or similar open source model as teacher model instead of much powerful Claude model.
>
> Regarding the data leakage concern: the potential distribution-matching advantage from seeing Qwen3-8B responses during SFT is **symmetric** between the Coupled and Decoupled variants, since both use the same Qwen3-8B responses as input. Therefore, it does not affect the relative comparison between them. Furthermore, during RL training the actor's distribution shifts away from its initial policy, so later-stage responses are out-of-distribution for both RMs. The stability advantage of IP-GRM (Figures 4–5) occurs precisely in these later stages, suggesting the benefit is not driven by distribution matching.
>
> To address the request for a weaker open-source teacher, we replace Claude-3.5-Sonnet with **Qwen3.5-27B**:
>
> | Model (Teacher) | Overall | Chat | Chat Hard | Safety | Reasoning |
> |-----------------|---------|------|-----------|--------|-----------|
> | GRM | 61.65 | 76.90 | 55.83 | 67.41 | 60.04 |
> | IP-GRM (Qwen3.5-27B) | 67.51 | 78.14 | 56.43 | 77.61 | 64.47 |
> | IP-GRM (Claude-3.5-Sonnet) | 68.22 | 78.93 | 56.48 | 78.22 | 65.67 |
>
> With the weaker teacher, IP-GRM still achieves comparable performance (RewardBench 67.51 vs 68.22). The same pattern holds in downstream RL:
>
> | Model | WB-A | WB-B | WB-C | WB-D | WB-E | WB-F | WB-all | CW-v3 |
> |-------|------|------|------|------|------|------|--------|-------|
> | Qwen3-8B | 83.7 | 81.7 | 83.6 | 82.7 | 83.2 | 83.8 | 83.0 | 70.7 |
> | w. GRM | 85.8 | 83.9 | 85.8 | 85.1 | 85.7 | 85.2 | 85.2 | 74.2 |
> | w. IP-GRM (Qwen3.5-27B) | 87.6 | 85.9 | 87.8 | 86.7 | 87.9 | 87.0 | 87.2 | 77.1 |
> | w. IP-GRM (Claude-3.5-Sonnet) | 87.7 | 86.9 | 87.3 | 88.5 | 88.2 | 87.1 | 87.6 | 77.8 |
>
> With Qwen3.5-27B as the teacher, IP-GRM achieves RewardBench overall 67.51 (vs 68.22 with Claude), WB-all 87.2 (vs 87.6), and CW-v3 77.1 (vs 77.8). This indicates that IP-GRM's effectiveness is robust to teacher quality and does not rely on a strong closed-source teacher.
>
> > **W3:** There are inconsistency in abbreviations which are used i.e., GenRM / GRM, IP-GenRM/IP-GRM.
>
> We apologize for the inconsistency. In the revised version, we will unify all abbreviations: **"GRM"** for the baseline coupled Generative Reward Model, and **"IP-GRM"** for our proposed Independent Principle GRM. All occurrences of "GenRM" and "IP-GenRM" will be replaced accordingly.

---

> > ### Author Rebuttal · Reviewer_mTjs · 2026-04-02
> >
> > Thank you for the additional ablations. Upon closer inspection of the RewardBench subcategory breakdown, the gains from decoupling appear to be disproportionately concentrated in the Safety subcategory. Comparing IP-GRM (Decoupled SFT) against GRM (Coupled SFT) safety improves +10.00, while others only show a marginal improvement. A similar pattern holds in the weaker teacher ablation. Could the authors explain why the decoupling strategy yields such uneven gains across subcategories? If principle drift is a general phenomenon affecting all evaluation dimensions, one would expect more consistent improvements rather than a single subcategory driving the overall number. Given that this core concern remains unresolved, I am maintaining my current rating.

---

> > > ### Author Response · Authors · 2026-04-04
> > >
> > > We thank the reviewer for the follow-up question. To address the concern regarding the distribution of improvements across subcategories, we provide additional benchmark evidence.
> > >
> > > > **Follow-up:** The gains from decoupling appear to be disproportionately concentrated in the Safety subcategory on RewardBench. If principle drift is a general phenomenon, one would expect more consistent improvements rather than a single subcategory driving the overall number.
> > >
> > > We acknowledge the observation regarding the original RewardBench (v1) results. To verify whether the concentration of gains in Safety is a general limitation or specific to that benchmark, we evaluate the models on three additional reward benchmarks:  RewardBench v2, RM-Bench, and RMB.
> > >
> > > 1. RewardBench v2.
> > >
> > > | Model | Factuality | Focus | Math | Precise IF | Safety | Ties | Overall |
> > > |-------|:--:|:--:|:--:|:--:|:--:|:--:|:--:|
> > > | GRM | 31.16 | 47.29 | 40.24 | 25.62 | 36.22 | 52.94 | 38.51 |
> > > | GRM (Coupled SFT) | 32.84 | 47.69 | 40.94 | 27.50 | 38.44 | 55.29 | 40.20 |
> > > | **IP-GRM (Decoupled SFT)** | **39.58** | **49.09** | **43.72** | **35.04** | **47.34** | **64.71** | **46.98** |
> > >
> > > Here the largest gains over GRM (Coupled SFT) come from Ties (+9.42), while Factuality, Precise IF and Safety is comparable.
> > >
> > > 2. RM-Bench.
> > >
> > > | Model | Chat | Code | Math | Safety | Overall |
> > > |-------|:--:|:--:|:--:|:--:|:--:|
> > > | GRM | 56.56 | 53.97 | 54.31 | 76.17 | 61.74 |
> > > | GRM (Coupled SFT) | 57.56 | 54.39 | 55.87 | 76.49 | 62.35 |
> > > | **IP-GRM (Decoupled SFT)** | **61.54** | **56.08** | **62.09** | **77.76** | **64.81** |
> > >
> > > Over GRM (Coupled SFT), the improvements are: Chat +3.98, Math +6.22, Code +1.69, while Safety is only +1.27.
> > >
> > > 3. RMB.
> > >
> > > | Model | Harmlessness | Helpfulness | Overall |
> > > |-------|:--:|:--:|:--:|
> > > | GRM | 36.30 | 44.14 | 41.02 |
> > > | GRM (Coupled SFT) | 36.81 | 44.62 | 41.59 |
> > > | **IP-GRM (Decoupled SFT)** | **38.87** | **46.54** | **43.85** |
> > >
> > > Over GRM (Coupled SFT), Harmlessness (+2.06) and Helpfulness (+1.92) show nearly identical gains.
> > >
> > > Across these diverse benchmarks, improvements are observed in all subcategories, and the specific subcategory benefiting the most varies. This indicates that principle drift is a general phenomenon, and the concentration on Safety is specific to the composition of RewardBench v1 rather than a limitation of the decoupling strategy.
> > >
> > > Additionally, GRM (Coupled SFT) yields only minor gains over the original GRM across all three new benchmarks, confirming that the performance improvements stem primarily from the decoupling strategy itself.
> > >
> > > We hope these additional evaluation results address your concern. We will incorporate these findings into the revised manuscript and look forward to your further feedback.

---

### Official Review · Reviewer_ELqx · 2026-03-10

**Soundness:** 3
**Presentation:** 3
**Significance:** 3
**Originality:** 2
**Overall Recommendation:** 4
**Confidence:** 4

**Summary:**

This paper identifies a failure mode in principle-based Generative Reward Models (GRMs): when evaluation principles are generated after observing the response (Q + R → P), the resulting rubric can be contaminated by response-level artifacts, a phenomenon the authors call "Principle Drift." This coupling creates a feedback loop during RL training where hacked behaviors (e.g., verbosity, self-justification) are rationalized by the reward model, leading to inflated scores. To address this, the authors propose IP-GRM, which decouples principle generation from response evaluation by first generating principles from the question alone (Q → P), then scoring the response against these fixed criteria (Q, P, R → J, r). They also introduce a Principle Cache that reuses a single principle set across all responses in a GRPO group, improving throughput by ~24% while ensuring evaluative consistency. Experiments on RewardBench and GRPO-based creative writing training show that IP-GRM suppresses reward hacking, stabilizes training dynamics, and achieves state-of-the-art performance among open-source models on WritingBench and Creative Writing V3.

**Compliance With Llm Reviewing Policy:**

Affirmed.

**Final Justification:**

The authors are able to answer my questions, and I'm happy to raise my score.

**Key Questions For Authors:**

1. Residual hacking via Stage 2: Have you tested whether the actor can still learn to hack the Stage 2 judge even with fixed principles? For example, by training for more steps or by explicitly crafting adversarial responses that target the judgment phase? A negative result here would significantly strengthen the paper; a positive result would be important to acknowledge.

2. Generalization beyond creative writing: Do you have preliminary results on other open-ended tasks (e.g., coding, open-ended QA, or dialogue)? The paper's claims about "removing a key pathway for reward hacking" are general, but the evidence is domain-specific. Even a small-scale experiment in a second domain would improve confidence in generalizability.

I am happy to raise my score if these two questions can be addressed properly.

**Limitations:**

The authors' discussion of limitations is minimal. The Impact Statement is generic ("none which we feel must be specifically highlighted here"), and the paper does not discuss several important limitations: (1) the residual attack surface at the judgment stage, (2) the narrow domain focus on creative writing, and (3) the reliance on LLM-based evaluation without human validation. A more thorough limitations section would strengthen the paper. The authors are encouraged to expand this discussion.

**Strengths And Weaknesses:**

### Strengths
1. Soundness. The controlled experiment on RewardBench (Figure 2) provides clean evidence for Principle Drift. The three-condition design (P⁺, P⁰, P⁻) isolates the effect of conditioning on R, and the fact that P⁻ anomalously inflates scores for rejected responses is particularly convincing. The downstream RL experiments are thorough: the authors show not only final benchmark numbers but also training dynamics (Figure 4), KL divergence trajectories (Figure 5), and capability retention on IFEval/TruthfulQA (Figure 6), which together paint a coherent picture of the failure mode and its resolution.

2. Presentation. The case study in Figure 1 is effective at communicating the core idea. The paper is generally decently written and well-structured, and the progression from motivation → formulation → training → inference is logical.

3. Significance. The paper addresses a real and practically important problem. Creative writing and other open-ended tasks are increasingly the frontier for RLHF, and reward hacking in these domains is a major bottleneck. The Principle Cache is a nice practical contribution that makes GRM-based RL more feasible. The fact that an 8B model trained with IP-GRM outperforms much larger open-source models (QwQ-32B, Qwen3-235B) on writing benchmarks is a strong result.

4. Originality. While the individual components (decoupled generation, rubric-based evaluation, caching) are not entirely novel, the specific identification of the Q+R→P coupling as a reward-hacking channel, and the clean fix via conditional independence, constitutes a useful conceptual contribution. The Principle Drift framing provides a concrete and testable hypothesis that future work can build on.

### Weaknesses
1. Notation introduced too late and used too early. The introduction makes heavy use of the shorthand Q, R, P (and expressions like Q + R → P, Q(Q → P)), before these variables are formally defined. A reader encountering the paper for the first time must infer the meaning from context. While Q (question) and R (response) are somewhat guessable, P (principles) is specific to this work and not standard notation. The paper would benefit from a brief forward definition early in the introduction (e.g., after the first mention), rather than deferring all formalism to Section 3.2.

2. The fix addresses one attack surface but leaves others open. IP-GRM prevents the response from influencing the rubric, but the Stage 2 judge is still an LLM reading the response. Persuasive self-justifications, strategic formatting, or other artifacts can still bias per-criterion scoring even under a fixed rubric. The paper does not test or discuss this residual vulnerability. Additionally, because the rubric is generated from Q alone, it may fail to anticipate response-level pathologies that a Q+R-conditioned rubric could catch. The paper's framing suggests the problem is largely solved, but the actual guarantee is narrower.

3. Limited domain evaluation. All RL experiments focus on creative writing. While this is well-motivated as a domain susceptible to reward hacking, the generalizability of the approach to other open-ended tasks (summarization, dialogue, code review, instruction following more broadly) is untested.

The RewardBench evaluation provides some breadth, but the improvements there are moderate. Additionally, reproducible details of the RewardBench evaluation are not provided. Compared with the RewardBench leaderboard (https://huggingface.co/spaces/allenai/reward-bench), several scores in the subcategories are not listed. Are the scores conducted under the same setting as the official RewardBench Leaderboard setting? If it is relevant, can you run more open source models under the same settings and report the results? If not, I suggest finding other domains where AIs are known to often reward hack (e.g., coding) and provide your results. As the paper positions itself to address the broad reward hacking phenomena, only presenting creative writing results is not sufficient to support its claim.

4. Dependence on teacher model quality. The SFT data is generated using Claude-3.5-Sonnet as the teacher model. The paper does not discuss how sensitive IP-GRM's performance is to the quality or biases of this teacher. If the teacher's Q-only principles are poorly calibrated or systematically miss certain evaluation dimensions, the student inherits these deficiencies. An ablation varying teacher quality or using a weaker teacher would strengthen the claims.

5. Evaluation methodology concerns. The final writing quality is assessed by GPT-4o-mini, which is itself an LLM judge susceptible to its own biases. It is unclear whether IP-GRM's improvements reflect genuine quality gains or better alignment with GPT-4o-mini's particular preferences. Using a second, independent judge (or human evaluation) would provide stronger evidence.

---

> ### Author Rebuttal · Authors · 2026-03-29
>
> We thank the reviewer for the thorough evaluation. We address each concern below.
>
> > **W1:** Notation introduced too late. — We agree and will add forward definitions at first occurrence in the revised version.
>
> > **W2 & Q1:** Have you tested whether the actor can still learn to hack the Stage 2 judge even with fixed principles?
>
> This is an important question. We extend GRPO training from 150 to 300 steps:
>
> | Step | 50 | 100 | 150 | 200 | 250 | 300 |
> |--|-|-|-|-|-|-----|
> | Training Reward (w. GRM) | 8.11 | 8.35 | 8.52 | 8.63 | 8.75 | 8.82 |
> | Training Reward (w. IP-GRM) | 7.06 | 7.30 | 7.47 | 7.49 | 7.48 | 7.49 |
> | WB-all (w. GRM) | 8.51 | 8.38 | 8.26 | 8.13 | 8.00 | 7.69 |
> | WB-all (w. IP-GRM) | 8.52 | 8.56 | 8.82 | 8.79 | 8.81 | 8.78 |
> | KL(×10⁻³) (w. GRM) | 0.65 | 0.76 | 0.71 | 0.74 | 0.61 | 0.98 |
> | KL(×10⁻³) (w. IP-GRM) | 0.11 | 0.14 | 0.16 | 0.15 | 0.17 | 0.19 |
>
> GRM's hacking worsens: training reward climbs (8.63→8.82) while true quality degrades (8.13→7.69). IP-GRM plateaus around step 150 with stable quality (8.78–8.81) and low KL (0.17–0.19). Over 300 steps, the probability of the actor learning to hack the Stage 2 judge is low. The fixed principles serve as an **anchor**—as in Figure 1, the judge assigns 3.2/10.0 on "Instruction Compliance" under a principle forbidding explanatory content.
>
> That said, we do not claim that IP-GRM eliminates all possible hacking pathways. We will expand the Limitations section to discuss the residual attack surface at Stage 2 and potential mitigations as future work.
>
> > **W3 & Q2:** Do you have preliminary results on other open-ended tasks?
>
> **1. OOD subcategories on RewardBench.** IP-GRM's SFT data is exclusively from creative writing, yet it improves on unrelated subcategories:
>
> | Model | math-prm | safety | coding | llmbar-natural |
> |------|--|-|-|--|
> | GRM | 53.91 | 67.41 | 63.90 | 72.34 |
> | IP-GRM | 57.91 | 78.22 | 68.66 | 77.66 |
>
> **2. Cross-domain RL on UltraFeedback.** We conduct GRPO on UltraFeedback (general instruction following, not creative writing):
>
> | Model | AlpacaEval Vanilla | AlpacaEval LC | Arena-Hard |
> |-------|-------------------|---------------|------------|
> | Qwen3-8B | 54.9 | 52.1 | 22.1 |
> | w. Scalar RM | 57.6 | 54.7 | 25.3 |
> | w. GRM | 59.1 | 54.3 | 26.8 |
> | w. IP-GRM | 64.4 | 57.6 | 31.4 |
>
> IP-GRM gains +5.3/+3.3/+4.6 over GRM, confirming the decoupling strategy generalizes beyond creative writing.
>
> > **W3 (cont.):** RewardBench evaluation details and reproducibility.
>
> We use a **pointwise** setting (scoring R+ and R- independently), matching GRPO's inference paradigm, which differs from the leaderboard's pairwise setting. Prompts follow Appendix C; all other settings follow standard protocol. Additional baselines:
>
> | Model | Overall | Chat | Chat Hard | Safety | Reasoning |
> |-------|---------|------|-----------|--------|-----------|
> | GRM-16B | 61.65 | 76.90 | 55.83 | 67.41 | 60.04 |
> | Skywork-Reward-V2 | 66.32 | 79.34 | 54.08 | 70.08 | 64.64 |
> | GRM-27B | 67.97 | 77.83 | 57.64 | 72.04 | 66.89 |
> | IP-GRM (Ours) | 68.22 | 78.93 | 56.48 | 78.22 | 65.67 |
>
> > **W4:** Dependence on teacher model quality.
>
> **Ablation 1: Coupled vs. Decoupled SFT** (same Claude teacher, same 8K prompts, same base model):
>
> | Model | RewardBench Overall | WB-all | CW-v3 |
> |--|---|--------|-------|
> | GRM | 61.65 | 85.2 | 74.2 |
> | GRM (Coupled SFT) | 63.11 | 85.6 | 75.1 |
> | IP-GRM (Decoupled SFT) | 68.22 | 87.6 | 77.8 |
>
> Coupled SFT improves RewardBench by +1.46; Decoupled SFT improves by +6.57. The primary improvement stems from the **decoupling strategy**, not additional SFT data.
>
> **Ablation 2: Weaker teacher.** Replacing Claude-3.5-Sonnet with Qwen3.5-27B:
>
> | Model (Teacher) | RewardBench Overall | WB-all | CW-v3 |
> |---|--------|--------|-------|
> | GRM | 61.65 | 85.2 | 74.2 |
> | IP-GRM (Qwen3.5-27B) | 67.51 | 87.2 | 77.1 |
> | IP-GRM (Claude-3.5-Sonnet) | 68.22 | 87.6 | 77.8 |
>
> Both outperform GRM, indicating IP-GRM's effectiveness is robust to teacher quality.
>
> > **W5:** Evaluation methodology concerns. GPT-4o-mini as judge.
>
> We re-evaluate with two additional judges:
>
> | RM (Judge) | WB-all | CW-v3 |
> |-|-|-|
> | Qwen3-8B (GPT-4o-mini)| 83.0 | 70.7 |
> | w. GRM (GPT-4o-mini) | 85.2 | 74.2 |
> | w. IP-GRM (GPT-4o-mini) | 87.6 | 77.8 |
> | Qwen3-8B (Claude-3.7-Sonnet)|77.3|61.3|
> | w. GRM (Claude-3.7-Sonnet) | 79.0 | 67.3 |
> | w. IP-GRM (Claude-3.7-Sonnet) | 81.6 | 72.3|
> | Qwen3-8B (Claude-Sonnet-4)|69.0|55.3|
> | w. GRM (Claude-Sonnet-4) | 70.2 | 61.8|
> | w. IP-GRM (Claude-Sonnet-4) | 72.8 | 68.0|
>
> The ranking **IP-GRM > GRM** is consistent across all judges, confirming genuine quality improvements.
>
> > **Limitations:** The discussion is minimal and generic.
>
> We will expand Limitations to discuss: (1) residual Stage 2 attack surface; (2) further do main validation beyond creative writing and instruction following; (3) reliance on LLM evaluation, with multi-judge checks as partial mitigation.

---

> > ### Author Rebuttal · Reviewer_ELqx · 2026-03-31
> >
> > I appreciate the authors' efforts in running the extra experiments, and most of my questions are addressed. I have raised my score.

---

> > > ### Author Response · Authors · 2026-04-01
> > >
> > > We sincerely thank the reviewer for taking the time to carefully read our rebuttal and for the positive reassessment. We are glad that the additional experiments have addressed all the concerns. We will incorporate all the discussed improvements in the revised version.

---

### Official Review · Reviewer_kXW7 · 2026-03-12

**Soundness:** 3
**Presentation:** 4
**Significance:** 2
**Originality:** 3
**Overall Recommendation:** 4
**Confidence:** 2

**Summary:**

This paper studies the problem of Principle Drift that may occur when Generative Reward Models (GRMs) evaluate generated reward. Existing GRMs typically generate reward scores and evaluation criteria simultaneously given a question and the response, which may cause the evaluation criteria to be influenced by the response itself, leading to Principle Drift. The paper empirically verifies the existence of such Principle Drift. To address this issue, the authors propose IP-GRM, whose core idea is to decouple the generation of evaluation criteria from the scoring process: evaluation criteria are first generated solely based on the question, and then the question, criteria, and response are used together for scoring, thereby avoiding the influence of the response on the evaluation criteria. In addition, the authors propose a Principle Cache mechanism, which allows generated principles to be shared across responses for the same question during reinforcement learning training, reducing computational cost while maintaining consistent evaluation standards. Experimental results show that the proposed method achieves better performance on benchmarks such as RewardBench, WritingBench, and Creative Writing V3, and demonstrates more stable reward signals and stronger robustness during reinforcement learning training.

**Compliance With Llm Reviewing Policy:**

Affirmed.

**Final Justification:**

I am not familiar with this field, and I will remain neutral

**Key Questions For Authors:**

The key questions mainly focus on the experiment in Figure 2:
1. How are the chosen response R+ and the rejected response R- obtained in the paper? Are they provided by the benchmark itself or generated by a certain zero-shot model or what?
2. Why do the experimental results in Figure 2 show that both positive and negative responses receive increased scores under both positive and negative principles P+/P-, rather than negative responses decreasing under positive principles? This requires further explanation and analysis.

**Limitations:**

YES

**Strengths And Weaknesses:**

Strengths:
1.	The formulation of the Principle Drift problem is novel and meaningful.
2.	The paper is clearly written and easy to follow.
3.	The method is straightforward but effective, and the experiments demonstrate its effectiveness.
Weaknesses:
1.	The experiment in Figure 2 lacks some core details. For example, it is unclear how the chosen response R+ and rejected response R- are obtained. These details are crucial for the proposal in this paper.
2.	The evaluation experiments only consider DeepSeekGRM-16B/Qwen3-8B, and lack validation on other models. This raises concerns about the generality of the method.

The key questions mainly focus on the experiment in Figure 2:
1. How are the chosen response R+ and the rejected response R- obtained in the paper? Are they provided by the benchmark itself or generated by a certain zero-shot model or what?
2. Why do the experimental results in Figure 2 show that both positive and negative responses receive increased scores under both positive and negative principles P+/P-, rather than negative responses decreasing under positive principles? This requires further explanation and analysis.

---

> ### Author Rebuttal · Authors · 2026-03-29
>
> We sincerely thank the reviewer for the constructive feedback and the positive assessment of our work. We address each concern below.
>
> > **W1 & Q1:** How are the chosen response R+ and the rejected response R- obtained in the paper? Are they provided by the benchmark itself or generated by a certain zero-shot model or what?
>
> Thank you for this question. As mentioned in Section 3.1 Motivation, the chosen response R+ and the rejected response R- are directly provided by the RewardBench benchmark itself, not generated by us. RewardBench is a standard evaluation benchmark for reward models, where each instance consists of a prompt Q paired with a chosen response R+ and a rejected response R-. These pairs are curated from multiple established sources.
>
> > **Q2:** Why do the experimental results in Figure 2 show that both positive and negative responses receive increased scores under both positive and negative principles P+/P-, rather than negative responses decreasing under positive principles?
>
> This is an insightful question and we provide a detailed explanation here. Principle Drift manifests as **criteria relaxation rather than directional bias**. When a GRM generates principles after observing a response (whether R+ or R-), it tends to produce criteria that rationalize the observed content—omitting dimensions where the response is weak, using vaguer constraint language, or shifting focus toward aspects the response handles well. These softened principles are more lenient for *any* response evaluated under them, hence all scores increase regardless of response quality. This is what enables reward hacking: the actor obtains high scores under relaxed criteria without genuinely improving quality, consistent with the reward–quality decoupling observed in Figure 4. Our IP-GRM avoids this by generating principles from Q only, keeping evaluation standards strict and response-invariant. We will include concrete examples of criteria relaxation under P+/P- in the appendix of the revised version.
>
> > **W2:** The evaluation experiments only consider DeepSeekGRM-16B/Qwen3-8B, and lack validation on other models. This raises concerns about the generality of the method.
>
> Thank you for raising this concern. We note that DeepSeek-GRM is currently one of the few well-recognized open-source GRMs with dedicated principle-based generative reward modeling capability, which is why we adopt it as our base model. In the current work, we provide validation across multiple dimensions: we demonstrate Principle Drift on DeepSeek-GRM-16B and 27B in Figure 2, and evaluate GRPO training with Qwen3-8B and Qwen3-8B-base in Table 2, both showing consistent improvements.
>
> To further assess generality, we conduct a new ablation replacing the teacher model used for SFT data construction. In the original paper we use Claude-3.5-Sonnet; here we replace it with Qwen3.5-27B, a weaker open-source model:
>
> | Teacher Model | Overall | Chat | Chat Hard | Safety | Reasoning |
> |---------------|---------|------|-----------|--------|-----------|
> | GRM  | 61.65 | 76.90 | 55.83 | 67.41 | 60.04 |
> | IP-GRM (Qwen3.5-27B) | 67.51 | 78.14 | 56.43 | 77.61 | 64.47 |
> | IP-GRM (Claude-3.5-Sonnet) | 68.22 | 78.93 | 56.48 | 78.22 | 65.67 |
>
> Downstream RL performance:
>
> | Model (Teacher Model) |  WB-A | WB-B | WB-C | WB-D | WB-E | WB-F | WB-all | CW-v3 |
> |----|------|------|------|------|------|------|--------|-------|
> | Qwen3-8B | 83.7 | 81.7 | 83.6 | 82.7 | 83.2 | 83.8 | 83.0 | 70.7 |
> | w.GRM | 85.8 | 83.9 | 85.8 | 85.1 | 85.7 | 85.2 | 85.2 | 74.2 |
> | w.IP-GRM (Qwen3.5-27B) | 87.6 | 85.9 | 87.8 | 86.7 | 87.9 | 87.0 | 87.2 | 77.1 |
> | w.IP-GRM (Claude-3.5-connet) | 87.7 | 86.9 | 87.3 | 88.5 | 88.2 | 87.1 | 87.6 | 77.8 |
>
> When replacing Claude-3.5-Sonnet with Qwen3.5-27B, IP-GRM's performance remains close. This indicates that IP-GRM's effectiveness stems from the decoupling strategy itself rather than from the specific teacher model

---

> > ### Author Rebuttal · Reviewer_kXW7 · 2026-04-01
> >
> > The response to weakness 2 is not convincing enough.

---

> > > ### Author Response · Authors · 2026-04-04
> > >
> > > We thank the reviewer for the follow-up. We understand the concern regarding generality and acknowledge that our previous response focused on teacher model variation, which did not directly address the core question of whether the method generalizes to different RM base models and actor models. We have now conducted two targeted experiments to address this directly.
> > >
> > > > **W2 (follow-up):** The response to weakness 2 is not convincing enough. The evaluation experiments only consider DeepSeekGRM-16B/Qwen3-8B, and lack validation on other models.
> > >
> > > We conduct full GRPO training using Qwen3-4B as the actor model:
> > >
> > > | Model | WB-A | WB-B | WB-C | WB-D | WB-E | WB-F | WB-all | CW-v3 |
> > > |-------|------|------|------|------|------|------|--------|-------|
> > > | Qwen3-4B | 79.1 | 77.9 | 79.6 | 79.6 | 80.2 | 81.9 | 79.5 | 64.7 |
> > > | w. Scalar RM | 81.3 | 79.5 | 81.3 | 81.4 | 81.6 | 81.9 | 81.1 | 68.2 |
> > > | w. GRM | 80.6 | 79.6 | 81.4 | 79.6 | 81.3 | 79.1 | 80.2 | 68.8 |
> > > | **w. IP-GRM** | **85.5** | **83.7** | **86.1** | **84.7** | **85.5** | **85.6** | **85.1** | **72.3** |
> > >
> > > IP-GRM achieves +4.9 on WB-all and +3.5 on CW-v3 over GRM, consistent with the gains observed on Qwen3-8B and Qwen3-8B-base .
> > >
> > > To demonstrate that the decoupling strategy is not tied to the DeepSeek-GRM, we apply our decoupled SFT procedure to Qwen3-4B as the RM base model and evaluate on RewardBench:
> > >
> > > | Model | Overall | Chat | Chat Hard | Safety | Reasoning |
> > > |-------|---------|------|-----------|--------|-----------|
> > > | Qwen3-4B | 59.47 | 74.53 | 50.28 | 59.57 | 58.25 |
> > > | IP-Qwen3-4B | **64.41** | **77.22** | **52.48** | **66.01** | **60.42** |
> > >
> > > Decoupled SFT increases overall accuracy by 4.94, including all subcategories. Unlike DeepSeek-GRM which is trained for reward modeling, Qwen3-4B is a general-purpose LLM. The effectiveness of the decoupling strategy on this non-specialized base model, along with the actor model experiments, demonstrates that our approach is generalizable and not specific to a particular model or architecture.
> > >
> > > We hope these additional ablation results address your concerns. We will incorporate all the discussed improvements in the revised version and look forward to your further feedback.

---

### Decision · Program_Chairs · 2026-04-30

**Decision:**

Accept (regular)

**Comment:**

Strengths: The paper introduces a novel, meaningful failure mode (“Principle Drift”) and a simple but effective way to address the failure (decoupling principle generation from scoring). Reviewers found it well written and supported by good empirical evidence, including better benchmark performance, more stable RL training, and a useful efficiency gain from Principle Cache.

Weaknesses: The main concerns were limited initial evaluation/generalization, especially beyond the first model/task setup; some missing experimental details and notation clarity; and questions about whether gains came from the decoupling method versus teacher/model choices. One reviewer also worried about residual judge hacking and reliance on LLM-based evaluation, though several of these concerns were partly or fully addressed during the author response.